
# Internal tides off the Amazon shelf during two contrasted seasons: Interactions with background circulation and SSH imprints

Michel Tchilibou[1], Ariane Koch-Larrouy[1], Simon Barbot[1], Florent Lyard[1], Yves Morel[1], Julien Jouanno[1], and Rosemary Morrow[1]

[1]LEGOS, Université de Toulouse, CNES, CNRS, IRD, UPS, Toulouse, France

**Correspondence:** Michel Tchilibou (michel.tchilibou@legos.obs-mip.fr)

**Abstract.**

The Amazon shelf break is a key region for internal tides (IT) generation. The region also shows a large seasonal variation of circulation and associated stratification. The objective of this study is to document how these variations will impact IT generation and propagation properties. A high-resolution regional model (1/36° horizontal resolution), explicitly resolving IT
is analyzed to investigate their interactions with the background circulation and stratification, over two seasons : first MAMJJ (March to July), with weaker mesoscale currents, shallower and stronger pycnocline, and second ASOND (August to December) with stronger mesoscale currents , deeper and weaker pycnocline. IT are generated on the shelf break between the 100 and 1800 m isobaths, with a maximum on average at about 10 km offshore. South of 2°N, the conversion from barotropic to baroclinic tide is more efficient in MAMJJ than in ASOND. At the eight main IT generations sites, the local dissipation is
higher in MAMJJ (30%) than in ASOND (22%). The remaining fraction propagates away from the generation sites and mainly dissipates locally every 90-120km. The remote dissipation increases slightly during ASOND and the coherent M2 fluxes seem blocked between 4°-6°N west of 47°W. Further analysis of 25 hours mean snapshots of the baroclinic flux shows deviation and branching of the IT when interacting with strong mesoscale and stratification. We evaluated sea surface height (SSH) frequency and wavenumber spectra for subtidal ($f < 1/28h^{-1}$), tidal $1/28h^{-1} < f < 1/11h^{-1}$) and super tidal ($f > 1/11h^{-1}$)
frequencies. Tidal frequencies explain most of the SSH variability for wavelengths between 300 km and 70 km. Below 70 km, the SSH is mainly incoherent and supertidal. The length scale at which the SSH becomes dominated by unbalanced IT was estimated to be around 250 km. Our results highlight the complexity of correctly predicting IT SSH in order to better observe mesoscale and submesoscale from existing and upcoming altmetrics missions, notably the Surface Water Ocean Topography (SWOT) mission.

**1 Introduction**

The passage of barotropic tidal currents over a sloping bottom or topographic feature in a stratified fluid generates internal waves that propagate at tidal frequency and are called internal tides or baroclinic tides. Internal tides induce (vertical) isopycnal displacements of up to tens of meters and are distributed into a set of vertical modes. The low-modes can propagate horizontally over hundreds to thousands of kilometers, carrying most of the generated baroclinic energy away from the internal





tide generation sites (Zhao et al., 2016). The higher mode internal tides waves are associated with high vertical shear and are prone to dissipate in the vicinity of the generation site (Zhao et al., 2016). The internal tidal currents can be several times larger than those of barotropic tides, with enhanced shear and bottom friction that will induce ocean mixing. For the highest modes (having shorter horizontal and vertical wavelengths), the breaking of internal tides results in an irreversible diapycnal mixing. When the mixing occurs at depth it impacts on the general overturning circulation (Armi, 1979; de Lavergne et al.,

2016; Laurent and Garrett, 2002; Munk and Wunsch, 1998), whereas when it is close to the surface, it can change the ocean surface temperature and salinity and thus impact on the air-sea fluxes and modify the local climate (Koch-Larrouy et al., 2010). Internal tides might play a key role in structuring the ecosystem in certain locations. Understanding where and how internal tides waves propagate and dissipate is a key issue that remains to be clarified.

The sea surface height (SSH) imprint of internal tides is a few centimeters, making them detectable by altimetry (Ray and

Mitchum, 1997). However, several years of altimetric SSH observations are needed to properly extract internal tide signals as they are aliased onto longer periods because of the satellite's temporal repeatability. It is possible to recover the internal tide signal from altimetric SSH by combining wavelength filtering and harmonic analysis (Ray and Zaron, 2016). Contrary to barotropic tides, which are extremely stable with time (except in some very particular locations), the baroclinic tides are permanently modulated by the background ocean variability. This modulation is linked to stratification variations at the internal

tides generation (Zilberman et al., 2011) and to interactions with the background circulation and its variability (eddies, currents) along the internal tides propagation pathways. Consequently, internal tide amplitudes and phases can be seen as the resulting sum of a "stable" component, called coherent tides, and a "variable with time" component, called incoherent tides. Of course, the definition of coherent and incoherent tides is closely linked with the time period considered: longer time periods will have a larger proportion of incoherent tides. The harmonic analysis of altimetric observations allows for the detection of coherent

internal tides, i.e. the internal tide component that is stable over the time of observation acquisition (Ray and Mitchum, 1996). Whereas incoherent internal tides with variable phase and amplitude are invisible to the harmonic analysis. Global maps of the major semi-diurnal (M2, S2) and diurnal (K1, O1) coherent internal tides have been constructed from multi-year and multi-satellite altimetry missions (Zhao et al., 2012; Zaron, 2019; Ray and Zaron, 2016; Kantha and Tierney, 1997). They show that altimetry observations are dominated by low mode internal tides that radiate from major ocean ridges (mid-Atlantic, Hawaiian,

western Indian), seamounts and continental shelf slope. Similar internal tide hotspots are obtained from harmonic analysis of ocean circulation model SSH (Arbic et al., 2012, 2010; Shriver et al., 2012).

The seasonality of the stratification induces a seasonality of the internal tides. Seasons with a shallow pycnocline coincide with the generation of high vertical modes, while a deeper pycnocline leads mostly to mode 1 internal tide generation (Tchilibou et al., 2020; Barbot et al., 2021). Idealized and, less often, realistic studies have looked at internal tides and current interactions.

Ponte and Klein (2015) highlight the phase shift and the dispersion of internal tides as they pass through an unstable jet structure. Dunphy and Lamb (2014) found that baroclinic eddies with diameters comparable to mode 1 length scale (the first internal radius of deformation), gradually disperse internal tide energy towards higher modes following the resonant triad wave-wave-vortex theory. Kelly and Lermusiaux (2016), Kelly et al. (2016) and Duda et al. (2018) investigated the effects of the Gulf Stream on internal tides. They show that the baroclinic current and associated strong horizontal density gradient



deflect the baroclinic flux (reflection, refraction) while the strong jets above the thermocline trap and advect internal tides. Very few studies are dedicated to internal tides in the northern Brazilian continental shelf, even though it is a hotspot for internal tide generation (Baines, 1982) and dynamics, given the complex circulation and stratification patterns of the region. In global maps, some similarity is observed between the SSH spatial pattern of the coherent mode 1 semidiurnal internal tides and the SSH incoherent internal tides map, both are maximum in the western Atlantic near the Brazilian shelf, the ratio of incoherent
tide to total semidiurnal internal tides being up to 50% there (Zaron, 2017).

The North Brazilian continental shelf is a shallow wide shelf extending off the Brazilian coast in the western tropical Atlantic. The shelf break occurs along the 100 m isobath (Figure 1). Temperature and salinity along the north Brazilian continental shelf vary under the influence of the freshwater discharge of the Amazon and Para Rivers, the trade winds, the North Brazil current (NBC), and the tidal forcing, primarily the semi-diurnal M2 (Geyer, 1995; Ruault et al., 2020). In boreal spring, (from March
to July, MAMJJ in the following), the Intertropical Convergence Zone (ITCZ) reaches its nearest equatorial position, the NBC is weaker and coastally trapped over the Brazilian shelf, the Amazon river discharge is higher, and the Amazon plume spreads across the entire shelf from about 2°S to 5°N and sometimes as far as the Caribbean region (Johns et al., 1998; Lentz and Limeburner, 1995; Lentz, 1995; Molleri et al., 2010). As a consequence, high temperatures and low salinity are observed in the surface layers (Neto and da Silva, 2014). A deep isothermal layer that contrasts with the shallow mixed layer of the Amazon
plume leads to the formation of barrier layers near the shelf break about 50 m thick (Silva et al., 2005). In boreal summer and fall, (from August to December, ASOND in the following), the ITCZ migrates to its northernmost position near 10°N, the NBC is broader and deeper, with flows reaching their maximum value within August-November periods. The Amazon river discharge decreases to its minimum in November-December. During this period the plume only extends 200-300 km in front of the Amazon river mouth, and is carried eastward to the central Equatorial Atlantic by the NBC retroflexion (NBCR) north of
5°N (Johns et al., 1998; Garzoli, 2004; Molleri et al., 2010). The continental shelf density stratification for this period is mainly determined by the temperature vertical distribution (Silva et al., 2005). A tongue of waters cooler than 27.5 °C, associated with a western extension of the Atlantic Cold Tongue, is present at the surface along and seaward of the continental shelf break south of 3-4°N (Neto and da Silva, 2014; Lentz and Limeburner, 1995; Ffield, 2005; Marin et al., 2009). This leads to vertical density structures that are very different between MAMJJ and ASOND, especially at the thermocline depth.

During its annual cycle, the NBC develops a double retroflection, first into the Equatorial Undercurrent (EUC) in winter/spring and second into the North Equatorial Countercurrent (NECC) at about 5°N - 8°N near 50°W (Didden and Schott, 1993). The most prominent mesoscale features observed along the northeastern Brazilian coast are the large anticyclonic NBC rings that detach from the NBC retroflexion (NBCR) and transport heat and salt from one hemisphere to another. Some eddies are present at subsurface with no surface signature (Fratantoni and Glickson, 2002; Barnier et al., 2001; Richardson et al., 1994;
Silva et al., 2009). Less persistent eddies within the NBCR and several cyclonic/anticyclonic vortices coming from the eastern tropical Atlantic increase the EKE. Overall the EKE seasonal cycle is very well correlated with that of the NBC (Aguedjou et al., 2019), EKE is lower in MAMJJ and higher in ASOND (see Aguedjou et al., 2019, figure 4d). So the typical ocean conditions that can be found in the region can be represented by two well marked "seasons", highly contrasting in stratification, surface currents and EKE.



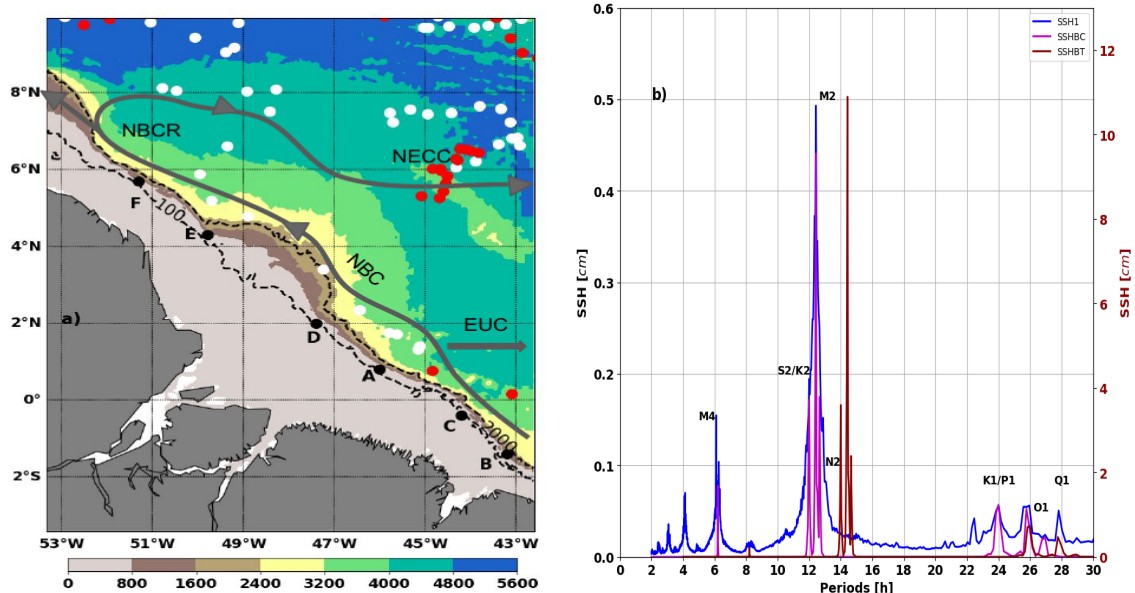

**Figure 1.** (a) Model bathymetry, Argo profiles locations during MAMJJ (white dot) and ASOND (red dot). Point A, B ,C,D,E and F are internal tides generation sites mentioned in Magalhaes et al. (2016). Dashed blacks contours are 100 m and 2000 m isobaths. Solid gray contours are NBC,NBCR and NECC pathways, EUC position is presented by a gray arrow. (b) SSH frequency spectra based on the 9.5 month (March to December) hourly signal of the coherent barotropic tides (SSHBT, brown), coherent baroclinic tides (SSHBC, magenta) and the residual between the full SSH and SSHBT (SSH1, blue). The brown spectrum refers to the right scale and is shifted by 2h for clarity. The spectra are averaged offshore of the 100m isobath.

The semidiurnal M2 accounts for about 70% of the barotropic tide crossing the North Brazilian continental shelf (Gabioux et al., 2005; Beardsley et al., 1995). Part of this barotropic energy converges to the Amazon river mouth (Geyer, 1995), another one induces a weakening of the mean currents on the shallowest part of the Amazon shelf and facilitates the offshore exportation of the plume by the NBC (Ruault et al., 2020). Internal tides are generated along the shelf break from several sites from A to E (Figure 1a) that have been primarily named in Magalhaes et al. (2016). From several sites A, B and F internal tides propagate
toward the open ocean. From C and D there is no evidence of their propagation. Magalhaes et al. (2016) suggests that at those sites most of the energy is dissipated locally which would explain why no energy left remains for the propagation. Intense semidiurnal Internal solitary waves (ISW, up to hundreds of kilometers from the shelf break) are consistently observed with SAR images propagating toward the open ocean (Magalhaes et al., 2016; Jackson, 2004). These ISWs are associated with the instability and energy loss of internal tides coming from A and B (Magalhaes et al., 2016; Ivanov et al., 1990). Modulation of
their propagation direction has been reported in Magalhaes et al. (2016), being rather south (45°) in Jul-Dec and rather north (30°) in Feb-May. The authors suggest that the stronger NECC in Jul-Dec (Figure 1a) might be a likely explanation for the ISW seasonal deviation.





Observations of internal tides are one of the objectives of the future wide-swath altimetry mission SWOT (Surface Water and Ocean Topography). SWOT aims to observe sea surface topography in 2D down to sub mesoscale of 15-40 km (Fu and Ferrari, 2008). As with Jason-class along track altimeter missions, SWOT is specifically designed to observe the major ocean tidal constituents. SWOT should provide the first 2D SSH observations of the generation, propagation and dissipation of internal tides, and their interaction with the finer-scale ocean circulation. Another objective of SWOT is to calculate the fine-scale geostrophically-balanced currents globally. Achieving this goal requires a highly accurate prediction and correction of the SSH fluctuations due to unbalanced motions, including barotropic tides and both coherent and incoherent internal tides. An accurate prediction of the incoherent internal tide from models or analysis remains a big challenge. So it is also important to understand what spatial scales of the ocean are impacted by the unbalanced incoherent internal tides, before calculating geostrophic currents. These scales can be estimated from SSH spectra, since the internal tides' small scale energy introduces spectral peaks that flatten the altimetric wavenumber spectral slope (Dufau et al., 2016; Tchilibou et al., 2018; Richman et al., 2012). The so-called "transition length scale" above which balanced motion dominates over unbalanced motions varies with latitude and eddy activity (Qiu et al., 2018). It becomes smaller in regions with high eddy kinetic energy, and can range from less than 50 km at mid to high latitudes, increasing to 100-250 km at low latitude (Savage et al., 2017; Qiu et al., 2018). One of the questions we wish to address in this study is the space-scales impacted by seasonal changes in the incoherent tides off the Brazil coast.

The other questions are about the internal tide characteristics during the two above-identified typical seasons. Are there strong seasonal variations in the generation, propagation and dissipation of the internal tide? What happens to the baroclinic flux after it passes through the stratification and circulation different from MAMJJ to ASOND? To answer these questions, we use a high-resolution model, presented in section 2, as well as in situ observations and the method of separating barotropic and baroclinic tides. In Section 3, we first validate the model and present stratification and EKE characteristics for the two seasons. Then, we describe internal tide energy budget terms, look at internal tide interactions with the current, evaluate the spectrum of the coherent and incoherent SSH over different frequencies and deduct the transition length scale. We summarize and discuss our results in section 4.

## 2 Data and method

### 2.1 Numerical model

The numerical model used in this study is NEMOv3.6 (Nucleus for European Modeling of the Ocean, Madec Gurvan et al., 2019). The model domain covers the Tropical Atlantic basin, and consists in a three-level, two-way embedding of : a 1/4° grid covering the Tropical Atlantic between 20°S and 20°N, a 1/12° grid covering the western part of the basin ($\sim$ 9km resolution, from 15°S to 15°N, 55°W to 30°W) and a 1/36° grid ($\sim$ 3 km resolution) covering the vicinity of the mouth of the Amazon (from 3.5°S to 10°N, from 53°W to 42.5°W, for more details see Ruault et al., 2020). All the three domains have 75 levels discretized on a $Z^*$ variable volume vertical coordinate, 24 of the levels are within the upper 100 m. They are coupled online via the AGRIF library in two-way mode (Blayo and Debreu, 1999; Debreu, 2000). A third-order upstream biased scheme (UP3)



with built-in diffusion is used for momentum advection. Laplacian isopycnal diffusion coefficients of 300, 100 and 45 $\mathrm{m^2\,s^{-1}}$
are used for tracer from the coarse to higher resolution grid. A time-splitting technique is used to solve the free surface, with the
barotropic part of the dynamical equations integrated explicitly. Atmospheric fluxes are from DFS5.2 (Dussin et al., 2016). The
Amazon river discharges are based on the interannual time series from the So-Hybam (2019) hydrological measurements. The

$1/4°$ model is forced at its open boundary by the tidal potential of the nine major tidal constituents (M2, S2, N2, K2, K1, O1,
Q1, P1, and M4) as defined by the global tidal atlas FES2012 (Finite Element Solution, Carrère et al., 2012). The $1/4°$ model
is initialized and forced at the lateral boundaries with daily velocity, temperature, salinity, and sea level from the MERCATOR
GLORYS2V4 ocean reanalysis (http://marine.copernicus.eu/documents/PUM/CMEMS-GLO-PUM001-025.pdf). The General
Bathymetric Chart of the Oceans (GEBCO) bathymetry (Weatherall et al., 2015) was interpolated on each of the three nested

grids. Figure 1a shows the domain and model bathymetry for the $1/36°$horizontal domain. Increasing the model horizontal
resolution from $1/4°$ to $1/36°$ leads to more intense and realistic barotropic tide energy conversion to baroclinic tides (Niwa
and Hibiya, 2011, 2014). The model was run over the period 2000–2015. In this study we concentrate our analysis on hourly
instantaneous output from the high resolution grid stored from 15/03/2015 to 31/12/2015. A twin configuration of the model
was run without the tidal forcing to allow spectral comparisons of the SSH with and without tides. More validations of the

model are available in Ruault et al. (2020).

## 2.2    Observations: Argo potential density and altimetric SSH

Model validation was performed by comparing model outputs with observations. The model potential density and stratification
were compared to the CORA (Coriolis Ocean Dataset for Reanalysis; Szekely et al., 2019) dataset. We benefited from the pre-
processing data done by Barbot et al. (2021) on CORA version 4.3 data to gather density profiles. CORA data were co-located

in time and space with model outputs. For 2015, most of the CORA data were ARGO float observations in our model area (see
location in Figure 1). AVISO gridded $1/4°$x $1/4°$x 7 day zonal and meridional geostrophic currents were used for the year 2015
to validate the model EKE. AVISO SSH and current anomalies are relative to 1992-2016 mean. Along-track 1Hz Saral/altika
sea level anomaly altimetric observations for the period 2013-2014 were used to validate model SSH wavenumber spectrum.
With its Ka-band, Saral altimeter has a lower noise level and gives access to smaller horizontal scales compared to Jason series

Ku-band altimeter (Verron et al., 2015). Altimetric data are all available on the website https://www.aviso.altimetry.fr. The
barotropic and coherent baroclinic SSH are validated by respectively comparison to FES2012 and to Ray and Zaron (2016)
internal tides SSH estimations based on altimetric observations.

## 2.3    Barotropic and baroclinic tide separation

The precise separation of the barotropic and baroclinic tide components is critical to achieve the internal tides diagnostics,

which in a first step requires us to clarify some basic definitions. Baines (1982) defined barotropic tides as the one present in
the absence of ocean stratification. Kunze et al. (2002) consider the barotropic tides as the depth-integrated tidal component in
a stratified ocean. Kelly et al. (2010) renewed Kunze et al. (2002) definition by adding a pressure depth-dependent correction
term to account for isopycnal heaving by free surface movements. These latter definitions lead to spurious barotropic energy





flux in the baroclinic energy flux (Nugroho, 2017, chap 6). Much better physical representation and baroclinic energy fluxes
are obtained by considering the barotropic tide as the fast mode (mode zero) in a Sturm-Liouville vertical mode solution, the
baroclinic tide being the sum of the non-zero modes (Kelly et al., 2012). The separation between barotropic and baroclinic tide
is further improved when the surface rigid lid condition commonly used when solving Sturm-Liouville equation is replaced by
a surface pressure condition based on the SSH free surface evolution (Tchilibou et al., 2018, 2020; Nugroho, 2017, chap 6) . In
the following, we will follow the latter definition and perform vertical mode decomposition using a free surface Sturm-Liouville
algorithm.

Prior to the vertical mode decomposition, the tidal constituents are extracted through a harmonic analysis, resulting in the
coherent components of currents, pressure, sea level (and energy flux) over the analyzed period. The barotropic/baroclinic
separation is processed for each of the nine tidal frequencies included in the simulation tidal forcing. For the purpose of SSH
analysis, we summed the baroclinic SSH tidal constituents into SSHBC referred as coherent baroclinic SSH. Similarly, SSHBT
was formed from the various barotropic SSH tidal constituents. The frequency spectra presented in Figure 1b for frequencies
higher than $1/24h^{-1}$ , confirms for the model that M2 is the main barotropic (SSHBT, brown curve) and baroclinic (SSHBC,
magenta curve) tide component in this part of the Atlantic. Our barotropic and baroclinic internal tide energy budget will
therefore concentrate on this constituent. Particular attention will be paid to the coherent and incoherent SSH in subsection 3.5
dedicated to SSH variability.

## 3   Results

### 3.1   Numerical tidal solution validation

We first evaluated the ability of the model to simulate the M2 coherent barotropic and baroclinic SSH. As mentioned in the
previous section, the harmonic analysis was applied to the model current and pressure (over the period from March to December
for the validation) and then the variables were projected onto the vertical modes to derive the barotropic and baroclinic SSH.
The amplitude and phase of the M2 model barotropic SSH were compared to those of FES2012 from Carrère et al. (2012),
which also forces the simulation at the lateral boundaries. The model correctly represents the propagation of the barotropic
tide (see Figure 3 of Ruault et al., 2020,  and Figure 2a and b here). The M2 barotropic tide is maximum near the northwest
and southeast of the Amazon mouth because of the landward propagation and convergence of the barotropic tide coming from
the open ocean. The simulated barotropic SSH is stronger on the shelf and slightly weaker in the open-ocean compared to
FES2012 but reproduces the same patterns. The differences on the shelf and the Amazon mouth might come from different
bathymetry and friction coefficient (see Le Bars et al., 2010, for sensitivity study) or difference in boundary conditions (closed
in our simulation whereas tide penetrates into the Amazon for FES2012). Comparing the model SSHBC with the filtered SSH
obtained from 20 years of altimetry observations (Figure 2c, 2d), the baroclinic tide surface signal is also well represented in
the model simulations. Internal tide SSH amplitudes reach 5 cm in front of the river mouth in both model and observations.
Internal tide SSH amplitude remains high along the 100m isobath over the whole area south of 2°N (including sites A and B)
in both the model and observations. North of 2°N, the two internal tide generation sites observed at the shelf break around 4°N



**Figure 2.** Top: M2 coherent barotropic SSH from (a) FES2012 (Carrère et al., 2012) and (b) the model. Bottom: M2 coherent baroclinic SSH from altimetry by Ray and Zaron (2016) and (d) the model. Amplitude is in color (unit: centimeters) and the phase in solid black contours. Dashed blacks contours are 100 m and 2000 m isobaths. Model are based on the 9.5 month hourly output

and 6°N (sites E and F, Figure 2c) are well represented in the model (Figure 2d), with similar offshore propagation. There are some differences between the model and the altimeter observations but the model generates and propagates similar internal tide patterns as in the real ocean. This good agreement is notable given that the hourly model simulation analysis is performed over a much shorter time period (9.5 months) compared to the Ray and Zaron (2016) empirical solution (over 20 years).





## 3.2 Validation of the simulated regional circulation: the contrast between MAMJJ and ASOND

In this subsection, we present the surface circulation, surface EKE and stratification characteristics associated with the MAMJJ and ASOND seasons described in the introduction from the different studies (Aguedjou et al., 2019; Didden and Schott, 1993; Silva et al., 2005; Neto and da Silva, 2014). MAMJJ and ASOND correspond to 1752 hours covering the periods shown
in Table 1. The MAMJJ shift of one week in August is necessary to have the same number of spring and neap tide cycles, which is better for the comparison of tidal harmonics. We will first check that the EKE and circulation conditions in the model are consistent with the observations. Then, as stratification is essential for internal tide generation and propagation, we will examine how it differs between the MAMJJ and ASOND seasons.

### 3.2.1 Mean current and EKE during MAMJJ and ASOND

First, 25-hour running means were performed to separate tide and high frequency from the low frequency mesoscale variability. Then EKE was evaluated using the anomaly of the 25-hour running mean current relative to the mean current over the entire period of availability of the hourly outputs (from March to December). As expected from the available literature, mean surface currents (Figure 3, arrows) are weak during MAMJJ, the NBC remaining trapped along the coast (Figure 3a). In ASOND, the NBC is wider and more intense, the NBC retroflection (NBCR) and the eastward current NECC are easily distinguished
(Figure 3b). The contrast between MAMJJ and ASOND is striking : the EKE is between 900-1200 $\mathrm{cm}^2\,\mathrm{s}^{-2}$ in MAMJJ, which is low compared to the values exceeding 2000 $\mathrm{cm}^2\,\mathrm{s}^{-2}$ along the NBCR/NECC pathways in ASOND. These EKE values agree with Aguedjou et al. (2019) who found larger diameter and more intense eddies in ASOND, but the generation and propagation of smaller (radius between 60-80 km) and lower SSH amplitude eddies (<3cm) during MAMJJ.

Figures 3c and 3d show EKE in MAMJJ and ASOND for the year 2015 from the AVISO data. They confirm the contrasts
revealed by our simulations offshore, although the simulations and altimetry observations are quite different mainly along the shelf break. The sources of these differences are multiple, including the horizontal resolution (1/4° for AVISO and 1/36° for NEMO), the reference period for the calculation of the mean current used to calculate the anomalies (1992-2016 for AVISO, 2015 for NEMO), the nature of the currents (geostrophic for AVISO, total for NEMO) and the processing of the altimeter signal at the limit of the continent which could be the reason why AVISO is maximum along the shelf break (Figure 3c and
3b).

### 3.2.2 MAMJJ and ASOND stratifications

About 50 Argo vertical profiles (see Figure 1 for locations), observing potential density to at least 1000 m in depth and with a stable Brunt Vaissala frequency (hereafter $N$), were selected over the 9.5 months of the study. They are spatially dispersed, with more than half of them during the MAMJJ period (Figure 1a).
The mean (red line) and standard deviation (red band) of Argo potential density and $N$ vertical profiles over the 9.5 month of simulation (Figure 4a and 4d), the MAMJJ (Figure 4b and 4e) and ASOND (Figure 4c and 4f) seasons are compared to the simulation (blue line and band) in Figure 4. The model and observations are collocated in time and space for better comparison.

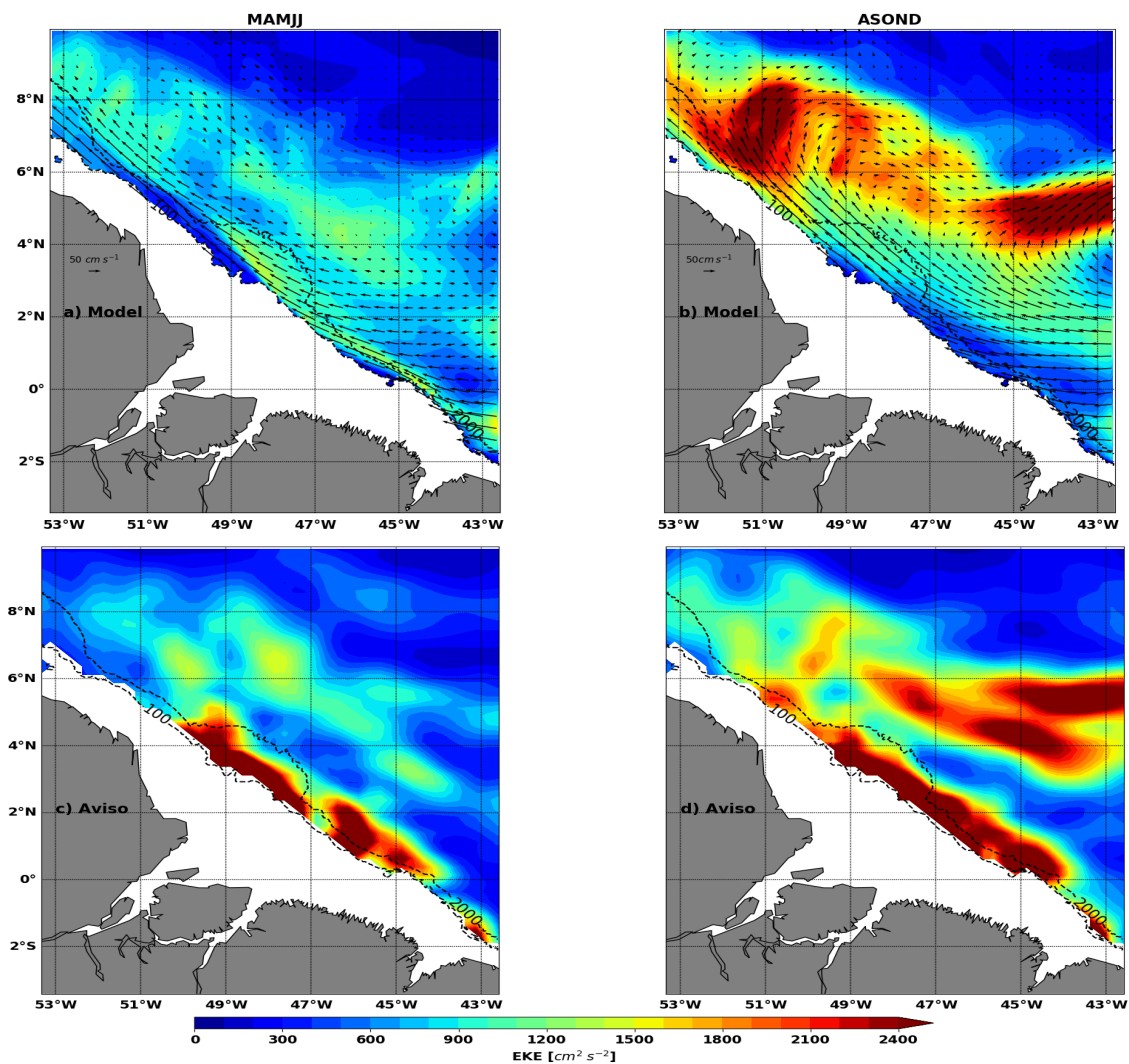

**Figure 3.** Top: model mean surface EKE (colors, units: $cm^2\,s^{-2}$) and current (arrows, units: $cm\,s^{-1}$) during (a) MAMJJ and (b) ASOND. Bottom: AVISO mean surface EKE during (c) MAMJJ and (d) ASOND. Dashed blacks contours are 100 m and 2000 m isobaths. Bathymetry less than 100m is masked

The Argo 2015 annual mean indicate a N maximum ($N_{max}$) around 100 m where the associated potential density is about 1025 $\mathrm{kg\,m^{-3}}$ (Figure 4a and 4d). The simulated N profile remains within the standard deviation of the ARGO profiles. The vertical

profiles for MAMJJ are almost identical to those of 2015 (Figure 4d and 4a).

The vertical profiles of N (Argo and model) are characterized by two maxima in ASOND (Figure 4c). The shallower is located in the first 50 meters of depth and is associated with very light water (Figure 4f), it is the signature of the Amazon plume extending eastward between August and October. The deeper maximum is associated with the pycnocline (Figure 4c and

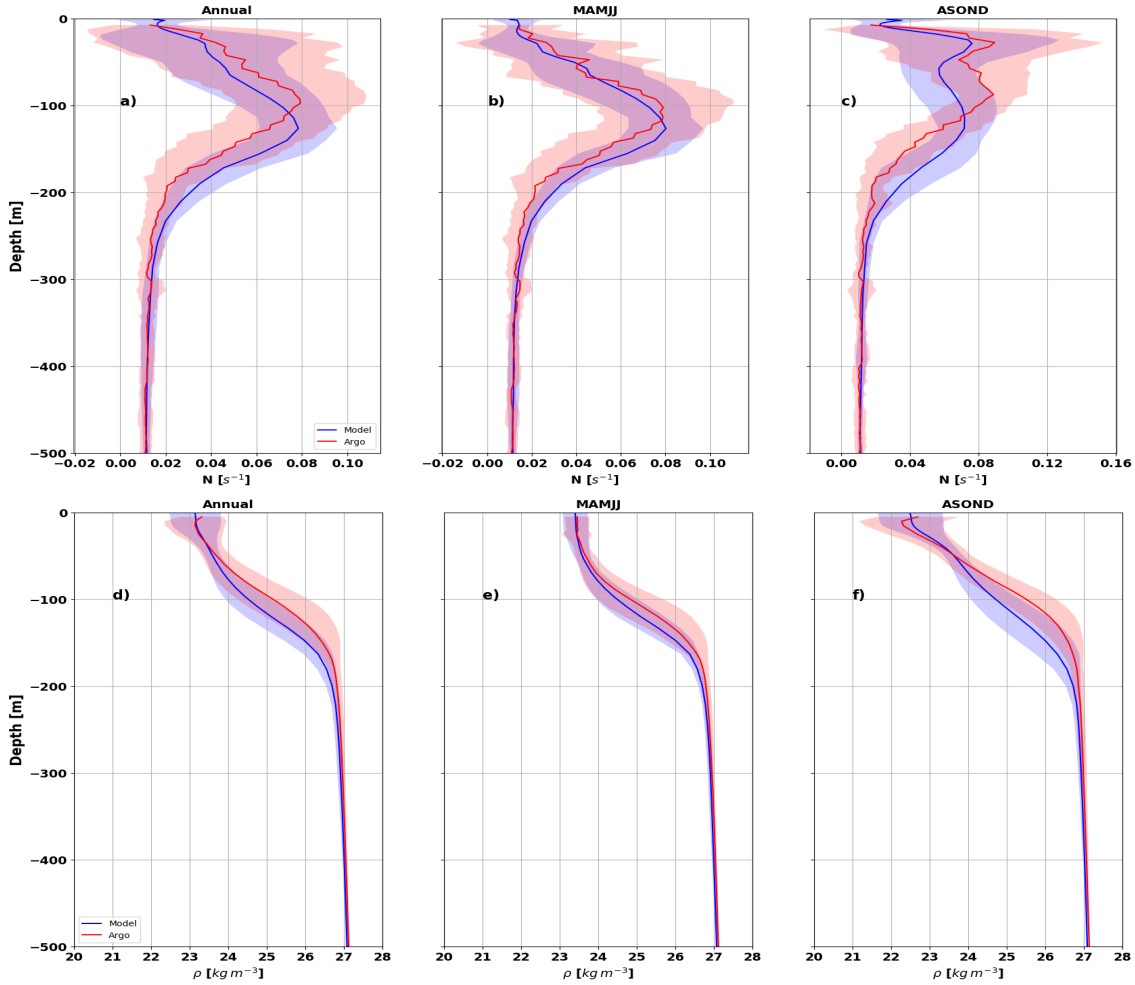

**Figure 4.** Mean vertical profiles from Argo (red) and model (blue) during (a,d) March to December 2015 (annual), (b,e) MAMJJ and (c,f) ASOND. Top: Brunt-vaissala frequency ($N$, units:$s^{-1}$ ). Bottom: Potential density (units: $kg\,m^{-3}$ ). The bands give the variability according to the standard deviation. See Figure 1 for Argo profiles location. Model and Argo are collocated in time and space

4e). In Barbot et al. (2021) idealized internal tide simulations based on the stratifications in this area, mode 1 and 2 baroclinic

SSH amplitudes increase linearly with the pycnocline depth. The mode 1 SSH wavelength also linearly increases with the pycnocline depth whereas the mode 2 SSH wavelength decreases. So, to further distinguish the stratification between the two seasons, we have evaluated the pycnocline depth based on the depth at which N is maximum (Figure 5 bottom) and N value at this depth (Figure 5 top). The first 50 meters of depth have not been taken into account to avoid the effects of the Amazon plume.

For all points $N_{max}$ value is higher in MAMJJ than ASOND (Figure 5a and b). Thus, a stronger higher mode internal tide generation is expected in MAMJJ. The depth of the pycnocline in MAMJJ is quite homogeneous in the area and around 130m







**Figure 5.** Top: $N_{max}$ value (units:$s^{-1}$) during MAMJJ (a) and (c) ASOND. Bottom: Pycnocline depth (depth of $N_{max}$, units: m) during MAMJJ (d) and (d) ASOND. The $N_{max}$ value and depth were deducted from the mean potential density over each season. Dashed blacks contours are 100 m and 2000 m isobaths. Bathymetry less than 50m is masked

(Figure 5c). In ASOND, the pycnocline deepens by more than 50 m and reaches 170 to 190 m in the area delimited by the NBC and its retroflection (Figure 5d). The internal tides generated in ASOND are expected to have less higher modes and a larger wavelength of the mode 1 (Barbot et al., 2021) . Also, the steep pycnocline slope established along the NBCR / NECC

(front) in ASOND, might act as a kind of barrier to the free propagation of the internal tide.





Table 1 summarizes the circulation and stratification characteristics between MAMJJ and ASOND. MAMJJ is the season of low current, low EKE, and a shallower, stronger pycnocline with weak spatial gradient. In ASOND, the currents are stronger, the retroflection is well developed, the EKE is strong and the pycnocline is deeper, weaker and with stronger horizontal gradient.

**Table 1.** Circulation and stratification characteristics during MAMJJ and ASOND seasons.

|  | MAMJJ | ASOND |
|---|---|---|
| Periods | 15/03/2015 - 07/08/2015 | 08/08/2015 - 31/12/2015 |
| EKE | Weak | High |
| NBC | Weak / Coastally trapped | High / Large |
| NECC / EUC / Retroflection | Weak | High |
| $N_{max}$ (Pycnocline) | Shallow / Strong / Low gradient | Deep / Weak / High gradient |

### 3.3   M2 coherent internal tide for MAMJJ and ASOND: Energy budget

By ignoring the energy tendency, the nonlinear advection and the forcing terms, the barotropic and baroclinic tide energy budget equations reduce to a balance between the conversion rate ($CVR$), the divergence of the energy flux and the dissipation (Buijsman et al., 2017; Tchilibou et al., 2020) as shown by the equations below.

$$div_h(F_{bt}) + D_{bt} + CVR = 0 \qquad \text{(W m}^{-2}\text{)}, \qquad (1)$$

$$div_h(F_{bc}) + D_{bc} - CVR = 0 \qquad \text{(W m}^{-2}\text{)}, \qquad (2)$$

with

$$CVR = grad_h(H)\overline{(U_{bt}P_{bc})_{z=H+\eta}} \qquad \text{(W m}^{-2}\text{)}, \qquad (3)$$

$$F_{bt} = \int_H^\eta \overline{(U_{bt}P_{bt})}dz \qquad \text{(W m}^{-1}\text{)}, \qquad (4)$$

$$F_{bc} = \int_H^\eta \overline{(U_{bc}P_{bc})}dz \qquad \text{(W m}^{-1}\text{)} \qquad (5)$$

In these equations, $bt$ and $bc$ indicate the barotropic and baroclinic tides, $U(u,v)$ is the horizontal velocity, $P$ is the pressure,

$F$ is the energy flux, $D$ is the dissipation term, $H$ is the bottom depth, $\eta$ the surface elevation, $grad_h$ and $div_h$ the horizontal gradient and divergence operators. The overbar indicates an average over a tidal period. $CVR$ appearing in the barotropic (Eq.1) and baroclinic (Eq.2) energy budget equations, determines the amount of barotropic tide energy converted into baroclinic tides. The baroclinic ($F_{bc}$, Eq.5) and barotropic ($F_{bt}$, Eq.4) flux respectively provide information on baroclinic and barotropic tides propagation pathways. We derived the dissipation $D$ from Eq.1 and Eq.2. Note that $D$ is more of a proxy of the real dissipation

because it may also include energy loss to other harmonics, non-linear terms and/or numerical dissipation (Nugroho et al., 2018).







**Figure 6.** Top: M2 conversion rate ($CVR$, color, units: $W\,m^{-2}$) and barotropic flux ($F_{bt}$, arrows, units: $W\,m^{-1}$ ). Middle: M2 dissipation (colors, $D$, units:$W\,m^{-2}$). Bottom: M2 baroclinic flux ($F_{bc}$, colors and arrows, units: $W\,m^{-1}$). Left column for MAMJJ (a,c,e) and right column for ASOND (b,d,f). Boxes are the hot spots of internal tide generation. Dashed blacks contours are 100 m and 2000 m isobaths. The black solid contours are parallels to the 100 m isobath drawn every 100 km and along which the integrations are performed for Figure 7.





**Table 2.** Location of boxes surrounding internal tides generation hot spots. In brackets, the color of the box as in Figure 6.

|  | Aa (Red) | Ab (White) | B (Green) | C (Cyan) | Da (Magenta) | Db (Yellow) | E (Blue) | F (Black) |
|---|---|---|---|---|---|---|---|---|
| lat (°N) | 0.85 / 0.3 | 1.4 / 0.85 | -1.15 / -1.75 | -0.1 / -0.65 | 1.95 / 1.4 | 2.55 / 2 | 4.55 / 4 | 6.05 / 5.5 |
| lon (°W) | 45.1 / 45.8 | 45.8 / 46.5 | 43 / 43.7 | 43.7 / 44.4 | 46.5 / 47.2 | 47.2 / 47.9 | 49.4 / 50.1 | 51.2 / 51.9 |

For MAMJJ (Figure 6a, vectors) and ASOND (Figure 6b, vectors), the barotropic energy fluxes are quasi-identical, as only a small fraction of barotropic energy loss is due to internal tide generation (compared to bottom friction) and the resulting change in the conversion rate is itself a small fraction of the total. The barotropic energy flux originates from the southeastern open
ocean and propagates towards the continental shelf. Initially directed towards the northwest, the fluxes gradually turn southward as they cross the shelf and converge towards the mouth of the Amazon River and Para River. The cross-shelf barotropic energy fluxes will be eroded through dissipation ($D_{bt}$, Table3) or through the generation of internal tides ($CVR$, Table3) according to Eq.1, until full extinction. North of 4°N in the NBC retroflection and NBC ring area, the barotropic tide flux decreases, likely because a large part was diverted toward the Amazon shelf.

Internal tide generation occurs along the shelf break (Figure 6a and 6b, negative blue color shading and $CVR$ in Table 3) between the 100 and 1000 m isobaths, with some exceptions until 1800 m (Figure 6a and 6b). Note that the positive conversion rate in Figures 6 and 7 (energy directed from the baroclinic towards the barotropic tides) can occur when the phase difference between the baroclinic bottom pressure perturbation and the barotropic vertical velocity exceeds 90° (Zilberman et al., 2011). Typically, this will happen at some distance of the generation site, at non-flat bottom locations, as the phase speed of the
baroclinic tides is much slower than the one of barotropic tides, making the phase difference vary quickly in the propagation direction. As noted in Figure 2, internal tide generation is stronger south of the Amazon cone (situated between 2-4°N/50°-47°W) than north of it.

For more detailed investigations, we divide the shelf break into 8 boxes of the same size as reported in Table 2 and plotted in Figure 6e. The hot spots of internal tide generations are located in A (Aa+Ab) and B sites (in good agreement with Magalhaes
et al., 2016), they respectively produce between 1.5 to 1.6 GW for A (Aa+Ab) and between 0.57 and 0.6 GW for B, depending on the season (MAMJJ or ASOND, Table 3). Sites C and Da also produce strong energy for internal tides (almost 0.4 GW, Table 3). Whereas the other sites show less conversion rate with about 0.3 GW for E, 0.2GW for Db and 0.1GW for F (Table 3).

In Table 3, we also calculate the ratio $P1$ (Eq.6), which can be seen as a proxy of the efficiency to convert internal tides from
the barotropic flux.

$$P1 = CVR/div_h(F_{bt}) \tag{6}$$

For $P1$ close to 1, internal tide generation explains most of the barotropic energy loss. If it is close to 0, then the divergence of the barotropic flux ($div_h(F_{bt})$) will be greater than the conversion rate, meaning that the barotropic flux exports most of the barotropic energy out of the box without local generation of internal tides. In the case of the A site, almost 80% of $div_h(F_{bt}$





**Table 3.** M2 baroclinic flux horizontal divergence ($div_h(F_{bt})$), barotropic dissipation ($D_{bt}$), convertion rate($CVR$), baroclinic flux horizontal divergence($div_h(F_{bc})$) and baroclinic dissipation ($D_{bc}$) at the hot spot of internal tide generation. Units are $GW$, boxes colors are in brackets. P1 is a proxy of the efficiency to convert internal tides from barotropic energy (Eq.6). P2 is a proxy of the efficiency of energy dissipated within the box (Eq.7). $CVR_{mode2}$ and $CVR_{mode3}$ are $CVR$ for vertical mode 2 and 3. Bathymetry less than 100m is masked.

| | | $div_h(F_{bt})$ | $D_{bt}$ | $CVR$ | $div_h(F_{bc})$ | $D_{bc}$ | $P1$ | $P2$ | $CVR_{mode2}$ | $CVR_{mode3}$ |
|---|---|---|---|---|---|---|---|---|---|---|
| Aa (Red) | ASOND | -1.15 | 0.21 | -0.95 | 0.78 | -0.17 | 0.82 | 0.18 | -0.19 | -0.03 |
| | MAMJJ | -1.07 | 0.16 | -0.91 | 0.66 | -0.25 | 0.85 | 0.27 | -0.24 | -0.06 |
| Ab (White) | ASOND | -0.81 | 0.17 | -0.64 | 0.51 | -0.14 | 0.79 | 0.21 | -0.17 | -0.02 |
| | MAMJJ | -0.67 | 0.09 | -0.57 | 0.42 | -0.16 | 0.86 | 0.27 | - 0.19 | -0.04 |
| B (Green) | ASOND | -0.99 | 0.43 | -0.56 | 0.46 | -0.1 | 0.56 | 0.17 | -0.08 | -0. |
| | MAMJJ | -0.98 | 0.38 | -0.6 | 0.43 | -0.17 | 0.61 | 0.29 | -0.16 | -0.02 |
| C (Cyan) | ASOND | -0.57 | 0.15 | -0.41 | 0.31 | -0.1 | 0.73 | 0.24 | -0.07 | 0. |
| | MAMJJ | -0.54 | 0.13 | -0.41 | 0.28 | -0.13 | 0.76 | 0.32 | -0.12 | 0.01 |
| Da (Magenta) | ASOND | -0.47 | 0.08 | -0.38 | 0.33 | -0.06 | 0.82 | 0.15 | 0.06 | -0.01 |
| | MAMJJ | -0.46 | 0.08 | -0.38 | 0.31 | -0.06 | 0.83 | 0.17 | 0.06 | -0.02 |
| Db (Yellow) | ASOND | -0.18 | -0.01 | -0.2 | 0.17 | -0.03 | 1.08 | 0.16 | 0.03 | -0.01 |
| | MAMJJ | -0.24 | 0.02 | -0.22 | 0.18 | -0.04 | 0.92 | 0.17 | 0.04 | -0.01 |
| E (Blue) | ASOND | -0.28 | -0. | -0.28 | 0.24 | -0.04 | 1.01 | 0.14 | 0.06 | -0.02 |
| | MAMJJ | -0.3 | -0. | -0.3 | 0.24 | -0.06 | 1.01 | 0.2 | 0.11 | -0.06 |
| F (Black) | ASOND | -0.07 | 0. | -0.07 | 0.05 | -0.02 | 0.94 | 0.22 | 0.03 | -0.01 |
| | MAMJJ | -0.1 | 0.02 | -0.09 | 0.07 | -0.02 | 0.82 | 0.2 | 0.05 | -0.02 |

is converted into internal tides, with only 20% flowing out of the shelf break in the Aa and Ab boxes. C and Da show similar behavior to A. In contrast, the B site has a smaller $P1$ ratio with 60% and less energy is converted into internal tides. Actually, B has the same $div_h(F_{bt})$ than A, but the efficiency to create internal tides is smaller (only 60%). This is due to the fact that the barotropic flux (Figure 6a and b) is perpendicular to the shelf break at the other sites (A, D, C, E and F), which is more efficient to create propagating internal tides, whereas the angle is smaller in the case of B. For Db and F sites, the $P1$ ratio is

even larger and close to 1. In this region north to 2°N (Db and F sites), the angle between the barotropic tides and the gradient of the topography is close to 90°, which is the most efficient angle for conversion of barotropic to baroclinic tides ($P1$ close to 1)

During MAMJJ the conversion rate $CVR$ in A (Aa+Ab) is slightly smaller (-7%) than in ASOND (MAMJJ : 0.91+0.57=1.48 vs ASOND : 0.95+0.64=1.59 GW, Table 3), whereas for B, Db, E and F, it is the opposite (between 5 to 10% higher in MAMJJ

than ASOND, Table 3). For C and Da the conversion rate remains identical between ASOND and MAMJJ. As shown in Table





3, the conversion efficiency ($P1$, Eq.6) is higher in MAMJJ than in ASOND for the sites A to Da south of 2° N. It is the reverse (or unchanged) for the northern sites Db to F. These changes might be due to the stratification changes from MAMJJ to ASOND and between south and north of 2°N. The higher efficiency to convert to internal tides south of 2°N in MAMJJ compared to ASOND is associated with the shallower and stronger stratification (Figure 5). Note that Eq.1 and 2 contains an
approximation since they do not take into account the nonlinear terms, also they might have some truncation errors due to large numbers. This might explain the larger numbers ($P1$>1) found for E and Db sites.

A proxy of the dissipation is given in Figures 6c and 6d as the residual between the conversion rate and the divergence of the baroclinic flux. Although it does not take into account non-linear terms, it is quite revealing of the coherent internal tides dissipation. At the generation sites, the conversion of internal tides ($CVR$ column, Table 3) is balanced by the export further
away through the baroclinic flux ($div_h(F_{bc}$ column, Table 3) and the local dissipation ($D_{bc}$ column, Table 3), following Eq.2. In regions away from generation sites, where $CVR$ equals zero, the dissipation explains all the loss of baroclinic energy.

Table 3 shows that dissipation is the highest for boxes A, B and C (between 0.1 and 0.3 GW), with the highest value for Aa. Smaller values of the dissipation are obtained at D, E, and F (between 0.02 and 0.06 GW). Regarding $div_h(F_{bc}$ , the largest values are for Aa (between 0.6 and 0.8 GW) while Ab and B have relatively smaller values (between 0.4 and 0.5 GW). The
divergence of the baroclinic flux gets smaller northward (about 0.3 to 0.2 GW for C, Da, Db and E) and is almost null for F. This is coherent with the baroclinic flux intensity (Figure 6e and 6f), where the flux exported toward the open ocean is decreasing from A to F. To discuss the dissipation, we defined the $P2$ ratio as follows :

$$P2 = D_{bc}/CVR \qquad (7)$$

$P2$ close to 1 means that internal tides generated in a box are dissipated locally there. On the contrary, if $P2$ is close to
0, the energy of the baroclinic tides propagates out of the box. As an example for site Aa (Table 3), during ASOND, $CVR$ = 0.95 GW and $div_h(F_{bc})$= 0.78 GW is exported away while 0.17 GW dissipates locally, yielding $P2$ = 0.18, so 18% of the internal tide energy generated in the box is locally dissipated. In fact, for the majority of the boxes, this ratio is between 15 to 30%, implying that 70 to 85% of baroclinic tide energy is radiated away. The largest $P2$ ratio occurs at C for both ASOND and MAMJJ (24% and 32% respectively), then, Aa (18% and 29%), Ab (21% and 27%), B (17% and 29%), F (22% and 20%),
E(14% and 22%) and Da (15% and 17%) and Db (16% and 17%). For all sites except F, $P2$ ratio is stronger in MAMJJ than ASOND, meaning that MAMJJ is more favorable to local dissipation. In the 8 boxes, the generation of mode 2 and 3 is larger in MAMJJ compared to ASOND (see CVR for mode 2 and 3 columns of Table 3), as expected for a season with shallower pycnocline Barbot et al. (2021). Once higher modes are generated, instabilities are more probable, and thus local dissipation is higher.

The conversion rate and the baroclinic dissipation in Figure 6 were integrated every 10 km along parallels to the 100m isobath (the shelf break reference) and presented as a function of distance from the shelf break in Figure 7. The conversion to internal tide occurring between the 100m and 1800m isobaths in Figures 6a and 6b result in a $CVR(CVR_{max})$ peak of -45 $Wm^{-2}$ at 10 km offshore (Figure 7a). There is no conversion away from the shelf break, $CVR$ varies very little between

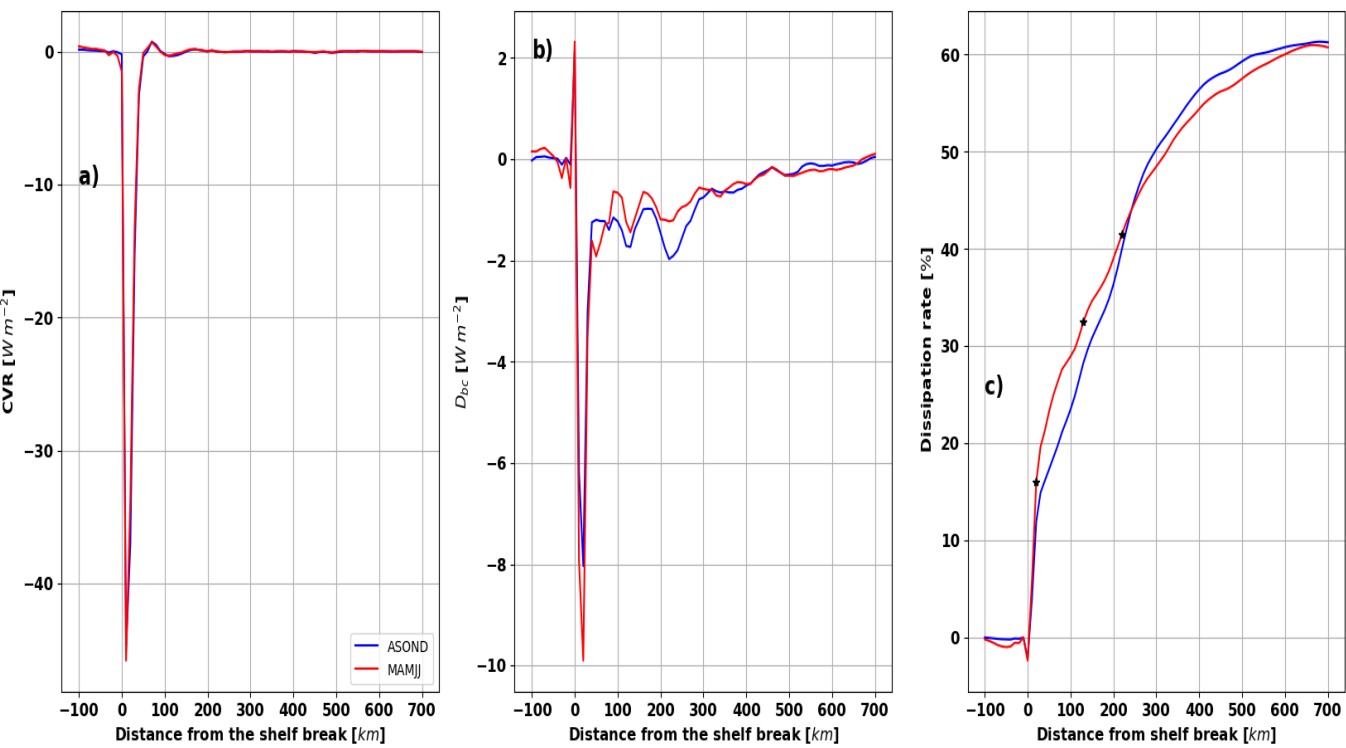

**Figure 7.** (a) $CVR$ (units: $W\,m^{-2}$), (b) $D_{bc}$ (units: $W\,m^{-2}$) and (c) dissipation rate (%) as a function of distance from the continental shelf break. $CVR$ and $D_{bc}$ are integrated every 10 km, the dissipation rate is the ratio between the cumulative sum of $D_{bc}$ and the sum of 0 to 50 km of $CVR$. ASOND in blue and MAMJJ in red. The black stars are the location of the three peaks of maximum dissipation.

MAMJJ and ASOND (Figure 7a). The maximum of $D_{bc}$ occurs a little bit offshore compare to the generation (20km), it is

separated from a second peak located 110 km offshore, between the second and the third peak there is 90 km (Figure 7b). It is relatively the same distances that separate the negative patches of dissipation in Figure 6c and 6d.

We defined the dissipation rate (Figure 7c) as a function of distance to the shelf break by dividing the cumulative sum of dissipation from Figure 7b by the sum between 0 and 50 km of $CVR$ from Figure 7a. The limit at 50 km was chosen because $CVR$ reaches zero for the first time around 50 km offshore. The 50 km distance can also be considered as the boundary

between local dissipation at the internal tide generation site and the remote dissipation. The dissipation rates at the distances associated with the three dissipation peaks (beams) (see star in Figure 7c) are 16%, 32%, and 41% during the MAMJJ, and 11%, 28%, and 40% during the ASOND. A significant increase in dissipation rate is observed between the second and third beams during ASOND compare to MAMJJ. At 50 km from the coast, the dissipation rates are 23% during MAMJJ and 17% during ASOND (Figure 7c), these dissipation rates express the local dissipation rate for the whole area. Furthermore, they

exactly match with the mean of P2 (Table 3) of the eight boxes surrounding the main internal tide generation sites in Figure 6, and confirm that local dissipation is stronger during MAMJJ (due to higher internal tide mode of shorter propagation). At 700





km, the dissipation rates are 60% during MAMJJ and 61% ASOND (Figure 7c), which allows us to deduce remote dissipation rates of 37% during MAMJJ and 44% during ASOND. Again, the remote dissipation is stronger during ASOND than during MAMJJ.

It is evident in Figure 6e and 6f that the M2 baroclinic tide does not propagate identically in ASOND and MAMJJ. During ASOND, the northward propagation of the coherent internal tide is stopped at about 300 km from the shelf break. During MAMJJ, the baroclinic flux from A reaches 8°N while the baroclinic flux from F and E have branches that extend further north. The cessation of northward propagation of the M2 coherent baroclinic flux in ASOND can be at first order associated with more remote dissipation (Figure 7c). However, the dissipation rates for the two seasons both converge to 60% in Figure

7c. Even if there is a 7% offset between the remote dissipation rate during ASOND and MAMJJ, there remains about 40% of the internal tide energy undissipated. The dissipation hypothesis is therefore not the best to explain the differences between the coherent baroclinic flux propagations. The discrepancies certainly reflect the modulation in time of the interactions between the internal tides and the bottom circulation.

### 3.4   Internal tides interactions with the background circulation

Generally, internal wave propagation and dissipation are modified in the presence of a horizontal density gradient or a background current. When passing through a baroclinic eddy, the internal tidal beam is subdivided into different divergent or convergent branches depending on how it enters the vortex or is deflected by its edge (Dunphy and Lamb, 2014). Strong currents such as the western boundary currents (Gulf stream, Kuroshio) are likely to trap, refract and reflect the internal tide flux (Duda et al., 2018). A horizontal density gradient and the corresponding pycnocline slope can have a mirror effect on the

internal wave and thus refract, reflect or at worst prevent its propagation (Li et al., 2019).

NBC, NBC retroflection (NBRC or NECC), and mesoscale eddies are associated with currents and frontal structures that can disrupt the propagation of the internal tide from the Amazon shelf break. This part of the work is a first approach to describe the variability of internal tides flux between ASOND and MAMJJ as observed above (Figure 6e and 6f). To investigate more precisely the tides/circulation interactions, we make the choice to leave aside the harmonic analysis approach,

which does not allow us to depict short term changes in the internal tide propagation characteristics. Instead, we make use of time filtering over a 25-hours period, which provides a fair separation of tidal and non-tidal processes, at the sacrifice of individual tidal constituents diagnostics, leaving the neap and spring tides modulation in the filtered tidal signal. In addition, the barotropic/baroclinic tides separation is performed by using (Kelly et al., 2010) method instead of vertical modes decomposition. This method, computationally less heavy than the vertical mode method, does not allow us to distinguish the vertical

modes, but such a distinction is not crucial for the present discussion.

Some selected snapshots of the 25-hour mean baroclinic flux, relative vorticity and currents are presented in Figure 8. As expected, the 25h mean eliminates the tidal signal in the currents while preserving the background and mesoscale circulation. The 25h-averaged internal tide flux (computed from the hourly low pass filtered simulated currents and pressure, then averaged over 25 hours) refers now to the total baroclinic flux. i.e it includes all the modeled baroclinic modes and tidal constituents.

Even though the internal tide signal is dominated by mode 1 of M2, the stronger modes 2 and 3 in MAMJJ could add smaller





**Figure 8.** Examples of 25h mean snapshots of depth integrated baroclinic flux (colors and arrows, left, units: $W\,m^{-1}$), relative vorticity along the 1025 $kg\,m^{-3}$ isopycnal (color, right, units: $s^{-1}$) and horizontal velocity along the 1025 $kg\,m^{-3}$ isopycnal (arrows, right, units: $m^{-1}$) during (a,b) 09/11/2015 spring tide during ASOND, (c,d) 09/21/2015 neap tide during ASOND and (e,f) 04/19/2015 spring tide during MAMJJ. Bathymetry less than 100m is masked.





scales to the baroclinic signal. For greater clarity and because several eddies have a reduced surface signature in this zone (Garraffo et al., 2003), we have chosen to represent the mean current and relative vorticity along the 1025 $kg\,m^{-3}$ isopycnal which is representative of the thermocline spatial and temporal variability in the area and crosses the cores of the main currents. The snapshots illustrate the modifications of internal tide trajectories during its propagation.

Internal tide trajectories during ASOND are perfectly illustrated by two snapshots on the 09/11 (Figure 8a et 8b) and 09/21/2015 (Figure 8c et 8d) corresponding respectively to the periods of spring and neap tides. During this season, the very intense currents delimit a frontal line with a steep pycnocline slope. Along the 1025 $kg\,m^{-3}$ isopycnal, we can also distinguish anticyclonic eddies that skim the coast (Figure 8b and 8d). The signature of these eddies is intensified in the upper ocean but they have a significant barotropic signature too. In both cases, internal tides generated at site F propagate in a ocean where there

exists strong shear and stratification variability, in particular the recirculation regions and the northern edge of the retroflection. The internal tide beam from F becomes very quickly incoherent during neap tide (Figure 8a). The deviation of internal tides flux is also evident on the beams from sites E and D (Figure 8a and 8c). At about 6°N, where the current and front are strong (Figure 8b and 8d), the flux from E splits into several branches, the main one being oriented towards the northeast. From D, internal tide flux splits in two. The branch close to Db is deviated to the northwest between 2°-4°N and then to the northeast

between 4°N-6°N, and finally joins the main branch from E. Internal tide beam leaving Da rapidly merge with internal tide flux from A. Internal tide beams from A are initially directed northeastward but lean eastward at the front and current maximum (near 4°N). The deviation is such that internal tides do not propagate far north of 8°N (Figure 8a). On 09/21/2015, a small part of it follows a thin branch that reaches 8°N (Figure 8c).

     Figures 8e and 8f on the 04/19/2015 are examples of a spring tide during MAMJJ. In this season, both the currents and fronts

are weaker (Figure 8f), therefore the baroclinic flux originating from A almost does not deviate from its initial trajectory and propagates to 8°N, which is the most striking difference with ASOND. But the marked propagation of the baroclinic flux from F and the branching of the one coming from E are also interesting contrasts.

     To conclude, if the neap-spring tidal cycle obviously plays a role in the internal tide flux intensity, there exist significant differences in internal tide trajectories and structure (branching and merging) between MAMJJ and ASOND. Our examples

show how this can be explained by refraction and dispersion due to current and stratification variability, in particular in the frontal region. The presence of strong eddies along the coast in ASOND is also a source of high variability influencing the propagation of internal tides.

     The examples discussed in Figure 8 give indications for the interpretation and understanding of the coherent baroclinic fluxes in Figure 6. The harmonic analysis captures trajectories that have the highest occurrences after internal tide interactions with

the background circulation. For instance, in MAMJJ, as the northern retroflection extension is reduced, the baroclinic flux from F is less impacted by the circulation. This limits the refraction and the beam is therefore more intense and visible in Figure 6e than in Figure 6f. More notably, while the baroclinic flux propagation originating from site E to A seems to stop before 6°N in ASOND, it extends further north during the weaker circulation conditions in MAMJJ. Internal tides are not dissipated as one could argue for ASOND, but the interaction of internal tides with the background circulation induce refraction and branching





and the variability of the circulation in this time period is such that on average there is no preferred direction of propagation beyond 6°N in ASOND.

### 3.5 Coherent and incoherent SSH for MAMJJ and ASOND

As mentioned in the introduction, altimetric observations of SSH include high-frequency unbalanced components from the barotropic tides and from the coherent and non-phase-locked (incoherent) internal tides. Global model estimates of the barotropic
tide are applied as a correction to altimetric SSH before the data are used for ocean circulation studies (eg FES2014, Lyard et al., 2021). New global coherent internal tide corrections are also becoming available (eg M2 SSH, Ray and Zaron, 2016). However any residual errors from these tide model corrections will remain in the altimetric SLA data and pollute the calculation of geostrophic currents. The incoherent component of internal tides also remains in the altimetric SLA. So it is important to understand what spatial and temporal scales are affected by these ageostrophic components, so that adequate filtering can be
applied to remove them for ocean circulation studies. This section addresses these scales for the Amazon region.

In the previous subsections, we have shown how the baroclinic flux and internal tide interactions with the circulation differ between MAMJJ and ASOND. These changes should naturally show up in the SSH. To study the SSH variations, the hourly SSH of the tidal model is split as indicated by the equations below:

$$SSH1 = SSH - SSHBT \qquad \text{(cm)} \qquad (8)$$


$$SSH2 = SSH1 - SSHBC \qquad \text{(cm)} \qquad (9)$$

where SSHBT and SSHBC are respectively the coherent barotropic and baroclinic SSH deducted for each season from the barotropic / baroclinic tides separation made from the projection on vertical modes (see subsection 2.3). SSHBT and SSHBC are the sum of all tide constituents by which the model has been forced. SSH1 corresponds to the usual processing of altimeter observations from which the barotropic tide correction is removed. The coherent part of internal tides is then removed from
SSH1 to obtain SSH2 (Eq.9). SSH1 and SSH2 both have similar low-frequency components. The high frequency signal in SSH2 is associated with spatio-temporal variations of internal tides and the Inertia Gravity Waves (IGW) spectra, the whole constitutes the incoherent SSH.

To study the spatio-temporal scales of the coherent and incoherent tides, spectral analyses are performed on SSHBC, SSH1 and SSH2. Before the FFT calculation, SSH is detrended and windowed with a Tukey 0.5 window, as previously done in
Tchilibou et al. (2020). The spectra are integrated over different frequency bands. We consider the "subtidal" as the periods above 28h ($f < 1/28h^{-1}$), the "tidal" as the periods between 28h and 11h ($1/28h^{-1} < f < 1/11h^{-1}$), and the "supertidal" as the periods below 11h ($f > 1/11h^{-1}$). The sensibility to these cutoff frequency bands was tested without major changes to our results. Finally, a separate analysis of the SSH variations of the model without tides revealed that fluctuations associated with high frequency atmospheric forcing can be neglected here (not shown).





### 3.5.1 Geographical distribution of the SSH temporal Root Mean Square (RMS) for different frequencies band

The geographical distributions of the temporal RMS of SSH1 deduced over all frequencies (full RMS, Figure 9a and 9b), tidal frequencies (Figure 9c and 9d) and supertidal frequencies (Figure 9e and 9f) are shown for the MAMJJ (Figure 9, left) and ASOND (Figure 9, right) seasons.

For both seasons the maximum variations of $SSH1$ occur north of 6°N and west of 48°W (Figure 9a and 9b) where the retroflection of the NBC takes place (Figure 3). Along the NBCR/NECC, the RMS is greater than 4 cm and the EKE is maximal (Figure 3). At first order, these maxima express the intraseasonal mesoscale variations of the SSH. This is confirmed by the map of the geographical distribution of the subtidal RMS (not shown), which is also maximal along the NBCR/NECC. However, the maximum of the full RMS is also due to $SSH1$ variations between 11h and 28h (tidal frequencies). In the area 4°-6°N/43°W-45°W for example, the full RMS is on average 5 cm in MAMJJ and 7 cm in ASOND while the tidal RMS is about 3 cm over the two seasons (Figure 9c and 9d). As shown in Figure 6e and 6f, the tidal RMS suggests wave propagation from internal tide generation sites. Some internal tidal beams are noticeable on the full RMS, especially in MAMJJ.

Figure 10 represents the coherent SSHBC (Figure 10a and 10b) and incoherent $SSH2$ (Figure 10c and 10d) signals at tidal frequencies. The structures in Figures 10a and 10b are reminiscent of those derived from the harmonic analysis in Figure 2d, and are in agreement with the M2 baroclinic flux in Figures 6e and 6f. During ASOND, the tidal RMS of SSHBC (Figure 10b) is almost zero north of 6°N while it is about 3 cm and has a well-marked wave structure for the incoherent $SSH2$ (Figure 10d). In MAMJJ, the incoherent signal corresponds to large structures co-located with the coherent internal tide beams (Figure 10a and 10c).

The RMS of $SSH1$ at super tidal frequencies reveals previously unsuspected propagation (Figure 9e and 9f). Waves whose main generation sites coincide with points A, B and E propagate from the shelf break towards the open sea. Judging from the number of beams in the 0-4°N / 45°W-43°W box (Figure 9f), the wavelength of these waves is less than 70 km. At 2°N, the RMS of the waves coming from A increases from 1 cm to more than 2 cm, the intensification continues until 6°N where there is a merger with the beam of waves coming from E (Figure 9e and 9f). By analyzing $SSHBC$ and $SSH2$ at super tidal frequencies (figures not shown), it appears that the super tidal signal is essentially incoherent.

The $SSH1$ subtidal, $SSHBC$ tidal coherent, $SSH2$ tidal incoherent and $SSH1$ supertidal RMS are averaged over the entire domain and reported in Table 4 ("mean" columns) for MAMJJ and ASOND. The subtidal signal is three to four times greater than the tidal one, which is almost twice the supertidal. At tidal frequencies, the incoherent signal is as important (or slightly more important in ASOND) than the coherent one. If we also take into account the fact that the super tidal signal is incoherent, then the amplitude of the incoherent signature on the SSH becomes greater than the coherent one, justifying the incoherence ratio of more than 0.5 noted by Zaron (2017, its Figure 8) in this region. The transition from MAMJJ to ASOND is marked by an increase in the subtidal RMS of about 1 cm (Table 4), but there is no significant change between seasons for the tidal (coherent and incoherent) and super tidal RMS values. Thus, the seasonal variability of stratification and the interactions of internal tides with the current impact mainly the geographical distribution of the SSH than its intensity.







**Figure 9.** Root means square (RMS) of $SSH1$ for (a,b) all frequencies (full), (c,d) tidal frequencies ($1/28h^{-1} < f < 1/11h^{-1}$), supertidal frequencies ($f > 1/11h^{-1}$) during MAMJJ (left) and ASOND (right). $SSH1$ is the residual between the SSH and the coherent barotropic SSH ($SSHBT$), see Eq.8 . Units: $cm$. Bathymetry less than 100m is masked.


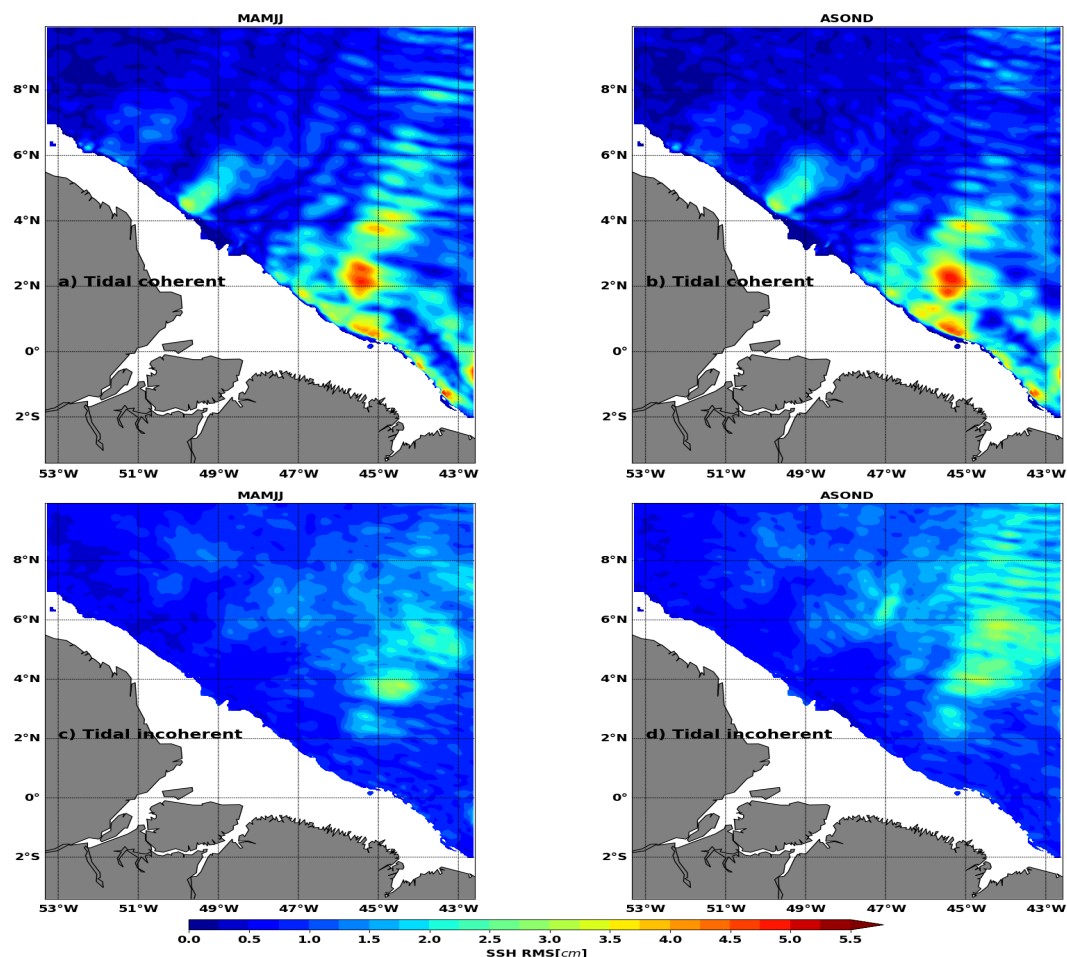

**Figure 10.** Root means square of (a,b) $SSHBC$ and (c,d) $SSH2$ for tidal frequencies during MAMJJ (left) and ASOND (right). $SSHBC$ is the coherent baroclinic SSH, $SSH2$ is the incoherent SSH defined as the residual between $SSH1$ and $SSHBC$, see Eq.9. Units:$cm$. Bathymetry less than 100m is masked.



**Table 4.** RMS of $SSH1$ at subtidal frequencies, coherent ($SSHBC$) and incoherent ($SSH2$) at tidal frequencies, and $SSH1$ at super tidal frequencies. Mean refer to the mean of RMS in Figure 9 and 10 over the model domain. Mode 1 and mode 2 refer to the RMS deducted from the integration of spectra in Figure 12 over the wavelength band 150-100$km$ and 100-70$km$ respectively

| RMS ($cm$) | Subtidal ($SSH1$) | Coherent tidal ($SSHBC$) | | | Incoherent tidal ($SSH2$) | | | Supertidal ($SSH1$) |
|---|---|---|---|---|---|---|---|---|
| | mean | mean | mode 1 | mode 2 | mean | mode 1 | mode2 | mean |
| MAMJJ | 3.68 | 1.06 | 1.52 | 0.61 | 1.04 | 1.1 | 0.96 | 0.62 |
| ASOND | 4.49 | 1.01 | 1.09 | 0.58 | 1.16 | 1.28 | 1.1 | 0.65 |

### 3.5.2 Meridional wavenumber-frequency spectrum

To better analyse the spatio-temporal scales impacted by the different components of the tides, meridional wavenumber-frequency spectra of $SSH1$ (Figure 11, right) and of hourly SSH of the model without tides ( NTSSH, Figure 11, left) are evaluated in the box 43°W-45°W/0°N-10°N through which a large part of the tidal and super-tidal $SSH1$ transit. The 10° latitudinal extension of the box limits the effects of overlap and flattening of the spectrum that would have occurred with a smaller latitudinal extension (Tchilibou et al., 2018). In Figure 11, negative wavelengths indicate southward propagation and positive wavelengths northward propagation. The MAMJJ and ASOND wavenumber-frequency spectra give similar conclusions, so only the MAMJJ spectra are presented. The energy distribution spectrum of the model without tides is asymmetric with a preference for northward propagations (Figure 11a). The energy maximum (in red, Figure 11a) is concentrated at subtidal frequencies (period >28h) and decreases as the frequency increases. In view of the amplitudes of spectrum with tides (Figure 11b), the injection of energy at high-frequency by the other oceanic mechanisms and atmospheric forcings can be considered negligible.

Not surprisingly, the introduction of tides into the model has boosted the high frequency energy, keeping the subtidal energy unchanged. Figure 11b shows clear maxima at diurnal ($0.042\ h^{-1}$, i.e. 12h), semidiurnal ($0.083 h^{-1}$, 12h) and higher harmonics (8h, 6h, 4h, 3h). At these different frequencies, the energy remains strong over a wide band of wavelengths while spreading out to neighbouring frequencies, and thus reflecting the mixing of coherent and incoherent internal tides. More generally, the frequency peaks mentioned above are not isolated but linearly connected to each other. Such a high-frequency distribution of energy in the spectrum is linked to the IGW field (Farrar and Durland, 2012), that contributes to both tidal and super tidal variations (Figure 11b).

### 3.5.3 Meridional wavenumber spectrum and transition scale

Satellite altimetry provides higher along track resolution of these SSH variations, with poorer temporal resolution. Here we consider how the full spatio-temporal structure of the model is projected onto wavenumber spectra. This method is often used to describe the spatial scales impacted by the ocean's turbulent energy cascade, and to identify scales impacted by the altimetric noise (Vergara et al., 2019; Xu and Fu, 2012; Chen and Qiu, 2021). The wavenumber-frequency spectra (Figure 11) have been time-integrated over selected frequency bands to obtain the meridional wavenumber spectrum in Figure 12. Wavenumber



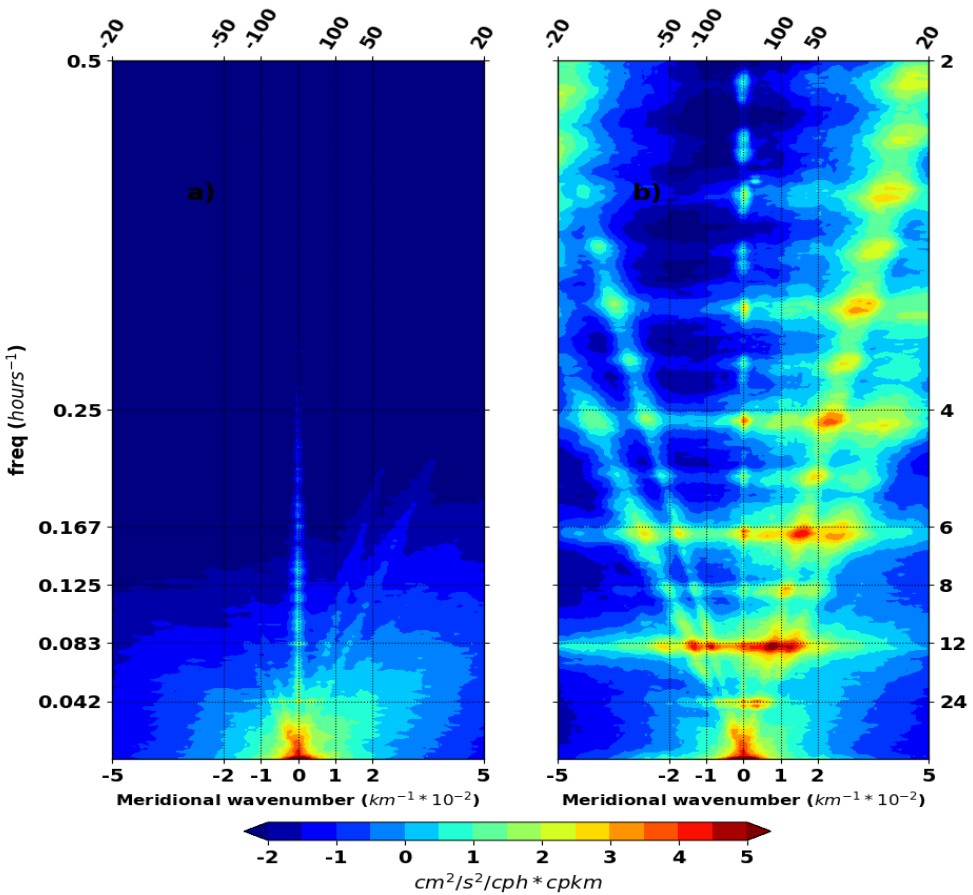

**Figure 11.** Meridional wavenumber-frequency of (a) the hourly SSH of the model without tide ($NTSSH$) and (b) the hourly $SSH1$ of the model with tide, both during MAMJJ. $SSH1$ is the residual between hourly total SSH and the coherent barotropic SSH. Spectra are evaluated within 0-10°N/43-45°W and averaged over the longitudes. Units: $cm^2\,s^{-2}/cph * cpkm$. Same results are obtained for ASOND.

spectra are shown for different SSH products with distinct integration frequency bands. For example, $SSH1\_full$ (in blue) is the spectrum of $SSH1$ integrated over all frequencies. The different frequency bands for SSH are also shown separately:

$SSH1\_subtidal$ (red), $SSH1\_tidal$ (green), $SSHBC\_full$ (magenta), $SSH2\_tidal$ (brown), $SSH2\_supertidal$ (cyan). For comparison purposes, we also present the wavenumber spectrum of the model with no tidal forcing, $NTSSH\_subtidal$ (No Tide SSH, orange) and the spectrum of the Saral/altika altimetry data, $Saral\_full$ (black). The two season spectra (MAMJJ and ASOND) are evaluated in box 43°W-45°W/0°N- 10°N where the subtidal, tidal and supertidal SSH have relatively high RMS (Figures 9 and 10).

The altimeter data ($Saral\_full$; black) and $SSH1\_full$ (blue) are both corrected for the barotropic tide only, and have similar characteristics (Figure 12a and 12b). In ASOND the two spectra overlap in the classical 250-70 km "mesoscale" wavelength band. Both the model and observations have a spectrum with an average k-1 slope in the 250-70 km band; they show



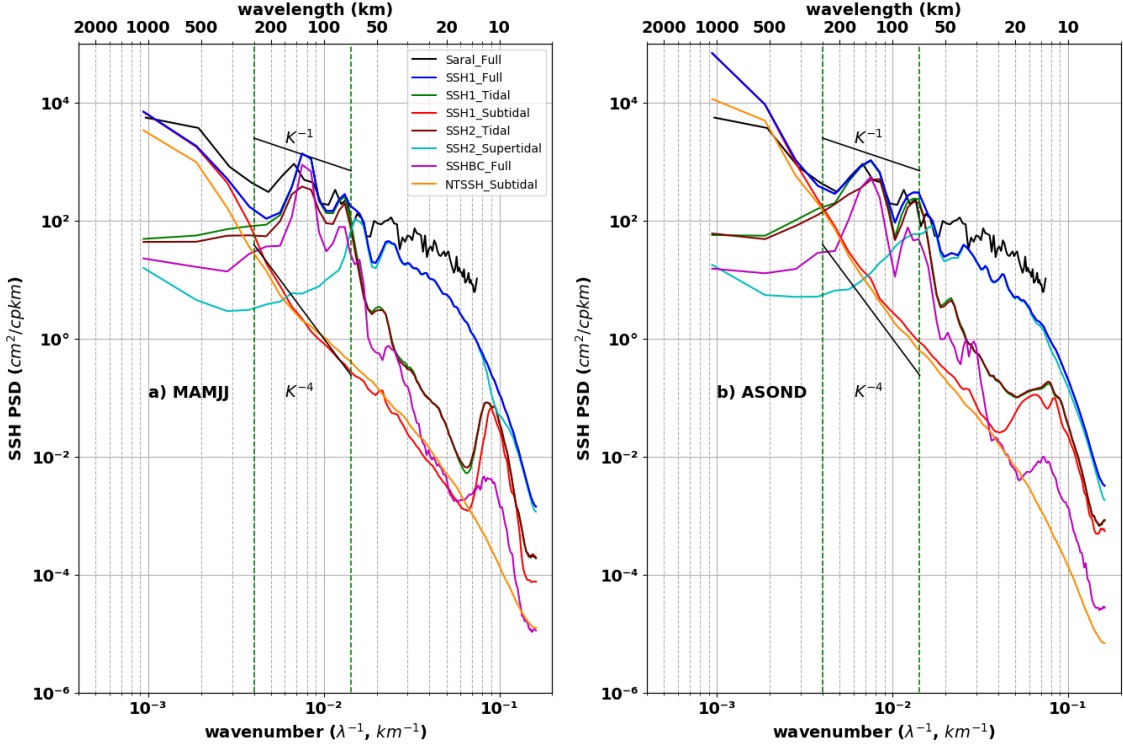

**Figure 12.** SSH Meridional wavenumber spectra during (a) MAMJJ and (b) ASOND. The spectra of the model are deduced by integrations on all frequencies (full), at subtidal frequencies (subtidal, $f < 1/28h^{-1}$), at tidal frequencies (tidal, $1/28h^{-1} f < 1/11h^{-1}$) and at supertidal frequencies (supertidal, $f > 1/11h^{-1}$) of meridional wavenumber-frequency spectra of $SSH1$ (hourly residual between the total SSH and the coherent barotropic SSH), $SSH2$ (hourly residual between the total SSH and $SSH1$), $SSHBC$ (hourly coherent baroclinic SSH) and $NTSSH$ (hourly SSH of the model without tide) meridional wavenumber-frequency spectra. Saral_full (in black) is the mean of Saral/Altika along track SSH spectra for the period 2013-2014. Spectra are evaluated within 0-10°N/43-45°W and averaged over the longitudes. The vertical dotted green line delimit the classical 250-70$km$ mesoscale band. Units are in $cm^{-2}/cpkm$.

peaks at mode 1 (120 km) and mode 2 (70 km) baroclinic wavelengths (same as $SSHBC\_full$). Despite the discrepancies at large scales and at scales smaller than 60km, the agreement between altimetry and model reinforces our confidence in the
model.

In Figure 12a and 12b, $SSH1\_subtidal$ (red) is closer to $SSH1\_full$ (blue) from 1000 to 300 km, from 300 to 70 km $SSH1\_full$ is dominated by $SSH1\_tidal$ (green), and $SSH2\_supertidal$ (cyan) explains the variations below 70 km. In the 250-70 km band, the RMS of the spectrum are 0.23 cm for $SSH1\_subtidal$ (red) and 0.21 cm for $NTSSH\_subtidal$ (orange) in MAMJJ, and 0.46 cm and 0.43 cm respectively in ASOND. In this wavelength band, the energy levels of the
$SSH1\_subtidal$ (red) and $NTSSH\_subtidal$ (orange) spectra are comparable in ASOND, their slope is in $K^{-4}$ (Figure





12b). In MAMJJ, a change in slopes is observed in both spectra (Figure 12a). Overall, the SSH at subtidal frequencies, i.e. the mesoscale imprints on SSH, does not significantly change when switching from the model without tide to the model with tides.

On the other hand, the RMS of $SSH1\_full$ (blue) and $SSH1\_tidal$ (green) are 2.46 cm and 2.4 cm in MAMJJ and 2.57 cm, 2.43 cm respectively in ASOND between 250-70 km. The closeness of the RMS of $SSH1\_full$ and $SSH1\_tidal$ and
their large deviation from $SSH1\_subtidal$ make the internal tides and IGWs the main contributors to the variation of SSH between 250-70 km. So for this example from 43-45°W off the Amazon shelf break, the SSH variations for meridional scales greater than 300 km are consistent with ocean circulation variations, the internal tides (and few IGW) of tidal frequencies dominate at scales between 300 and 70 km, whereas IGW (and few internal tides) of super tidal frequencies dominate at scales smaller than 70 km. This distribution according to the wavelengths is in agreement with Figures 6 and 7, and Table 3.

For both seasons, there is more energy in the tidal incoherent SSH than in the coherent internal tide ($SSH2\_tidal$ in brown versus $SSHBC\_full$ in magenta, Figure 12) at large (>300 km) and small scales (<60 km). As we suspected from Figure 10, the incoherent tidal and coherent internal tide have peaks at both mode 1 (120km) and mode 2 (70km) baroclinic scales. More precisely, integrating the spectra between 150-100km for mode 1 and 100-60km for mode 2 (Table4), leads to a stronger incoherent SSH than the coherent SSH for mode 1 in ASOND, and stronger incoherent tide for mode 2 over both seasons.

Savage et al. (2017) define the transition scale between balanced and unbalanced motion as the wavelength at which the amplitude of the spectrum at super tidal frequencies exceeds that of the subtidal frequencies. In our case, the $SSH2\_supertidal$ (cyan) and $SSH1\_subtidal$ (red) intersect around 152 km in MAMJJ and 133 km in ASOND (Figure 12, Table 5), i.e. at spatial scales close to those noted in the equatorial Pacific by Savage et al. (2017) . We also note a slight seasonal variation in the transition scale, which decreases by 20 km between MAMJJ and ASOND. However, defining the transition scale from the
super tidal is delicate in this tropical region where tidal variations are very strong. In fact, up to 67 km in MAMJJ and 62 km in ASOND (Table 5 ), the $SSH2\_supertidal$ (in cyan, Figure 12) is weaker than $SSH2\_tidal$ (brown, Figure 12). Instead, if we consider the transition scale to be defined between the subtidal (red) and the tidal incoherent (brown) or $SSH1\_tidal$ (green), then it becomes 250 km for both seasons (Figure 12, Table 5). If we are more interested in the coherent SSH (magenta) then the transition scales are 250 km in MAMJJ and 200 km in ASOND, the seasonality here is due to the change in EKE, the
stronger the EKE the smaller the transition scale. In any case, the transition between the balanced and the unbalanced occurs at scales beyond 150 km once we are no longer concerned with the super tidal. The application of the geostrophic approximation is compromised for meridian spatial scales below 250 km in this region.

**Table 5.** Transition lenght scale between balanced and unbalanced motion.

|  | Subtidal / Supertidal | Incohérent tidal / Supertidal | Subtidal / Incohérent tidal | Subtidal / Coherent |
|---|---|---|---|---|
| MAMJJ | 152 $km$ | 67 $km$ | 250 $km$ | 250 $km$ |
| ASOND | 133 $km$ | 62 $km$ | 250 $km$ | 200 $km$ |



## 4 Summary and Discussions

One of the challenges for the future SWOT mission is to propose appropriate processing to filter out most of the internal tides signals in the SSH products. Such an objective requires a clearer knowledge of internal tide dynamics including their temporal variability in various regions of the ocean. This study focuses on internal tides off the Amazon shelf, their interactions with the background circulation (currents and stratification) and their SSH signature during two strongly contrasted seasons. The analyses are based on 9.5 months (March to December 2015) of hourly outputs of a high resolution (1/36°) NEMO numerical model forced by explicit tides, that we validated by comparison with Argo and altimetry observations. The oceanic region off the Amazon shelf is strongly influenced by the seasonal cycle of the ITCZ, the Amazon River discharge, and the western boundary current NBC and its retroflection. Their combined actions give rise to strong contrasts in circulation and stratification. On the basis of the precise examination of the NBC and EKE cycles, the model simulations have been equally distributed between the MAMJJ (corresponding to March to July) and the ASOND (from August to December) seasons. In MAMJJ, the pycnocline is closer to the surface, slightly stronger and quite horizontally homogenous over the model domain, the currents and mesoscale activity are weak. In ASOND, the pycnocline is deeper (up to 50 m difference with MAMJJ), slightly weaker but with strong horizontal gradient along the NBCR/NECC path, the currents and mesoscale activity are intense. For each of the two seasons, the tidal frequency components have been separated by harmonic analysis and the projection on vertical modes is used to separate the barotropic tide from the baroclinic tide, in a way similar to Tchilibou et al. (2020). Harmonic analysis was also used to distinguish the coherent internal tide from the incoherent (we define as coherent the fields deduced from current and pressure harmonics, the incoherent being the residual between the total and the coherent fields for frequencies faster than 28 $h^{-1}$). The dominant tide component for this region is M2, so the M2 coherent barotropic and baroclinic flux, as well as the baroclinic dissipation and SSH have been detailed for the MAMJJ and ASOND contrasted conditions.

Whatever the considered season, the M2 barotropic tidal energy fluxes originate in the open ocean and reach the continental shelf. A large part of the barotropic tide crosses the shelf-break and converges towards the mouth of the Amazon River. The other part converts into baroclinic tides along the shelf break between the 100 and 1800 m isobaths. Eight poles of internal tides generation have been identified (see Table 1 and Figure 6 for position), the mains modeled sites are located at A and B (in good agreement with Magalhaes et al., 2016) where the conversion rate from barotropic to baroclinic tides was found to be around 1.5 GW (Aa+Ab, table 3) and 0.6 GW respectively. The conversion rate at the other sites are 0.4 GW for C and Da, 0.3 GW for E, 0.2 GW for Db and 0.1 GW for F (Table 3). These differences are explained by differences in the barotropic flux intensity and the angle between the flux and the topography slope. In the case of the A and C site, almost 80% of the barotropic energy is converted into internal tides, and only 20% will flow out of the shelf break. In contrast, B, which has a similar barotropic flux as A, is less efficient in generating internal tides since the angle is smaller than 90°. At the eight main internal tides generation sites, the conversion rate varies slightly from 5 to 10% between the two seasons. During MAMJJ the conversion rate in site A is slightly smaller than in ASOND, whereas for B, Db, E and F, it is the opposite. South of 2°N, the conversion from barotropic to baroclinic tide is more efficient in MAMJJ than in ASOND. Larger conversion rate might be due to the stratification modulation from MAMJJ (shallower and slightly stronger stratification) compared to ASOND. This





is in good agreement with idealized simulations of Barbot et al. (2021), that show that shallower stratification enhanced the conversion rate in this area.

Regarding dissipation of internal tides and its variability, we found that between 15 to 35% of internal tide energy dissipates locally near the eight generation sites, implying that 65 to 85% is radiated away. The largest local dissipation is found at C for both ASOND and MAMJJ (24% and 32% respectively), while Aa, Ab and B ratio are slightly smaller (17% and 29%), F (22% for both seasons), E (14% and 22%) and Da and Db are the smallest (15%, 17%). Local dissipation at the generation sites is higher in MAMJJ than in ASOND. MAMJJ has a shallower and slightly stronger stratification compared to ASOND which produces stronger higher baroclinic modes (mode 2 and 3). This makes the internal wave packet more unstable, and more

prompte for local dissipation. Offshore, internal tide dissipation hotspots have been observed along the propagating beams, with a distance separating them of about 90km to 120km, in good agreement with previous simulations (Buijsman et al. 2016). The distance (90 km) smaller than a mode 1 baroclinic wavelength (120 km) suggest that the dissipation would occurs in the model in the water column between 100 and 500m. It is also possible that the 90 km distance is a consequence of a change in stratification and particularly in the depth of the pycnocline as discussed by Barbot et al. (2021).

One of the most striking differences between MAMJJ and ASOND occurs when comparing the M2 coherent baroclinic fluxes (Figure 6). In ASOND, the offshore propagation of the baroclinic flux is like stopped after 200 km. In MAMJJ, on the other hand, the baroclinic flux propagates further away from the shelf break, the one coming from A reaches 8°N. As M2 fluxes are computed from a harmonic analysis, this "disappearing" M2 energy fluxes can be due either to true energy loss from the baroclinic tides, or to an increasing incoherent regime. We integrated the M2 conversion rate and dissipation every 10 km and

plotted them as a function of distance from the shelf break. The conversion to baroclinic tide is maximum 10 km offshore. The maximum of dissipation occurs at 20 km, two other peak of dissipation are observed offshore. The distance between the dissipation peaks is in agreement with the dissipation hotspots beams evoked above. The dissipation rate was estimated by dividing the cumulative sum of dissipation by the sum of 0-50 km $CVR$. The 50 km limit was set because of the first $CVR$ curve crossing at zero. In this study, 50 km is also the boundary distance between the local dissipation and the remote

dissipation independently of the generation site. During MAMJJ, the local and remote dissipation rates are 23% and 37% respectively, they change to 17% and 44% during ASOND. There is a 7% increase in remote dissipation during the ASOND, but for both seasons there is still nearly 40% of internal tide energy undissipated. Thus, energy loss through dissipation cannot explain all the discrepancies found between the propagation of the MAMJJ and ASOND coherent baroclinic flux, we show in this paper that the second hypothesis about the increase in the incoherent regime is more likely.

Indeed, snapshots of the total baroclinic flux averaged every 25 h are analyzed to investigate further the varying baroclinic flux of the two seasons. They reveal branching of the baroclinic flux at the level of the NBCR/NECC front, and possibly deviation by the NBC and coastal eddies-like structures. We associated the branching of the baroclinic flux with refraction, in good agreement with previous academic studies (Duda et al., 2018). We found that internal tide interactions with the background circulation depend on the spring/neap tide cycle and seasonal variations in the background circulation (the NBCR/NECC front

intensities). The baroclinic flux from F interacts with the background circulation just after it is generated. This explains why it propagates such a short distance off the shelf, compared to the others. The baroclinic flux generated at D, splits into two main





branches, the former merges offshore with the new branch resulting from the separation of the baroclinic flux from E while the second joins very quickly the baroclinic flux propagating from A. The baroclinic flux off A, although the most intense, undergoes an eastward deviation and sometimes branching at the front level (around 4°N). The baroclinic flux that appears
to stop and dissipate in ASOND are in fact rendered incoherent by the intensification of internal tide interactions with the background circulation.

An analyze of the geographical distribution of the SSH RMS and SSH wavenumber spectrum in different frequency bands complete these first results. We defined the subtidal band as periods greater than 28 h, the tidal band as periods between 28 h and 11 h, and the supertidal band as periods less than 11 h. SSH at tidal and super tidal frequencies is related to a mixture
of internal tides and inertail gravity waves. We found SSH RMS of 2 to 6 cm for the tidal frequency band and up to 2 cm for the supertidal one. In this tropical Atlantic region, tidal SSH is dominant at wavelengths between 250 km and 70 km while the super tidal SSH dominates for wavelengths below 70 km. The meridional wavenumber spectrum of the tidal coherent and incoherent SSH are characterized by peaks around 150-100 km and 100-70 km respectively associated with mode 1 and mode 2. At mode 1 wavelength, the peak of the incoherent tidal SSH spectrum is stronger than the coherent tidal in ASOND, the
order is reversed in MAMJJ. As it would be expected, the tidal incoherent SSH remains greater than the coherent signal at mode 2 wavelength both for MAMJJ and ASOND seasons. Using the Savage et al. (2017) criterion, the transition scale at which the SSH signal of geostrophic flow can be masked by unbalanced –wave- signature is around 150 km during ASOND and 130 km during MAMJJ. However, if we compare the energy levels of the subtidal and tidal spectra, then the transition scale is shifted toward 250 km for the two seasons (Table 5).
The contrast observed between ASOND and MAMJJ coherent baroclinic fluxes thus shows that the structure of the coherent signal, generated by the barotropic tide, is dependent on the interaction between internal tides and the background circulation/stratification over the analyzed time period. This result raises questions about the prediction of coherent internal tides which, to be optimal, must take into account variations in circulation and stratification. Internal tide trajectory patterns exhibiting several branches (Figure 7c for instance) are not retained by the harmonic analysis and are generally attributed to
incoherent tide. We have seen that beams originating from different generation sites (A and D for instance) can merge in some time periods. Possibly, it is this offshore merging of beams of various origins that sometimes gives the impression that there are only two internal tides generation sites on the Amazon shelf, as in Magalhaes et al. (2016) interpretation of SAR observations. SWOT will allow a far better description of the mesoscale activity, in particular for boundary currents. This better description of the seasonal and spatial variability, will improve our understanding of internal tides propagation and refraction around the
circulation. The spatial extension of the model does not yet allow us to make a clear statement on the eastward deviation of the baroclinic flux around 4°N east of 45°W. It could be a refraction or an advection. In any case it seems that at the front, the effects of stratification (refraction, reflection) compete with the effects of the current (advection) to define internal tides trajectory pattern. It would be interesting to quantify the respective impacts of stratification versus current. More investigations are needed to confirm the possible coastal deviations of internal tides by the NBC and the coastal eddy structures.
The SSH results highlight the risk of overestimating the RMS associated with the mesoscale circulation, without prior high frequency filtering. The predicted standard deviations of the measurement error uncorrelated to the instruments and in the case





where 15 km wavelength filtering is not applied to the SWOT observations are 2.74 cm for the raw data on 1 km x km grids and 1.35 cm in the case of 2 km x 2 km (Chelton et al., 2019): These noise levels are comparable to the SSH RMS at super tidal to tidal frequencies. There is therefore some high frequency physical signal that will be contained in SWOT noise. The

coherent baroclinic flux and the 25 h mean baroclinic flux snapshots presented in this study (Figure 7) are unanimous on the seaward propagation of the internal tide. On the other hand, the wavenumber-frequency (Figure 11b) shows that there is also southward propagation in the model. This southward propagations may be due to internal tides and IGWs reflection as they interact with the circulation or the topography. However, it is also possible that this reflection is indicative of a numerical tidal damping/radiating issue at the northern open boundary of the model. Finally, some of the wavenumber spectra in Figure 12 are

characterized by a hump at scales smaller than 20 km. We did not pay particular attention to this hump at 20 km which is close to the model effective resolutions.

In the past decade, many investigations have been motivated by the internal tide surface signature corrections for all altimetry missions but especially for the future wide swath altimetry SWOT mission. Various empirical atlases for surface internal tides have been derived from nearly 30 years of multi-mission altimetry, which reveal the coherent part of this signal over the

altimetry era. The altimetry community's more pressing issue is the non-coherent part that is left aside in these atlases, whose magnitude and variability are the main concerns today as they will significantly contribute to the SWOT error budget. Our investigations are a contribution to their quantification in a specific area, and demonstrate the large variability of the internal tide dynamics at seasonal timescales. They also suggest even higher variability if considering shorter timescales because of the interaction with the ocean upper circulation, indicating clearly that the internal tide correction will be one of the most

challenging problems for future altimetry data processing. In tropical regions with high seasonal variability, it is possible that internal tidal predictions at seasonal frequencies are more effective for altimetry data correction than annual prediction maps as currently proposed.





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
