# Peer review of "Internal tides off the Amazon shelf during two contrasted seasons: Interactions with background circulation and SSH imprints"

_Ocean Science, 2021_

## Author Comment (AC1)

We thank the reviewer for taking the time to review our manuscrip. We particularly appreciate the recommended publications on the internal tide regime. We are glad that the reviewer finds the article publishable after minor revisions. In the following, the reviewers' comments are in red and our responses in black colors.

1- The Introduction Section is well written and suited for the paper's results and discussions. Nonetheless, the issues related with IW propagation through a variable background (i.e. mesoscale variability via stratification and shear) have been discussed before in a few papers. Three papers are highlighted here which merit some additional framing and discussion in light of their previous results and the new (and valuable) insights provided by this paper. Nash et al. (2012) http://dx.doi.org/10.5670/oceanog.2012.44) Jeans and Sherwin (2001) https://doi.org/10.1016/S0278-4343(01)00026-7). Jensen et al. (2019) https://doi.org/10.1016/j.dsr2.2019.104710

We thank the reviewer for sharing these publications with us. Nash et al., 2012 was cited in the introduction (L40-45, and also L57). Instead of Jensen et al. (2019) https://doi.org/10.1016/j.dsr2.2019.104710, we discuss our results in regards to Jensen et al., (2018) which focus more on the interactions between internal tide and eddies: "This is not surprising since the interactions between internal waves and eddies can enhance the forward energy cascade (Barkan et al., 2021; Thomas and Daniel, 2021) or stimulate the generation of sub-mesoscale (Jensen et al., 2018).", (L584-586). The impact of internal waves on salinity and temperature is currently being analyzed, a dedicated paper to this study is in preparation. It will be the occasion to compare our results to those of Jensen et al., (2019).

2- Figs. 4c and 5d need some additional clarification. Stratification along the waves' typical propagation paths have two maxima between August and December. It may be misleading to assume that the IT energy is propagating in the same fashion as that of a waveguide with a single pycnocline (and any dynamics thereof). That does not hinder the results of the paper nor its conclusions, but it is better to clarify and discuss (very briefly!) that different dynamics are expected as reported in previous studie e.g. see the works by Theo Gerkema (https://doi.org/10.5194/npg-10-397-2003).

As recommended by the reviewer, we briefly discuss the double pycnocline and justify our choice to focus only on the deeper one: '' The presence of this near surface Nmax will have an impact on the modal structure of the internal tide and certainly impacts on the internal wave regime according to Gerkema (2003). We do not address the issue of the internal wave regime in this study. Vertical sections (Not shown), indicate that the internal tide interacts first with the base of the pycnocline around the depth of the second peak of N . Thus, to differentiate MAMJJ from ASOND, and following Barbot et al. (2021), we will use the deeper Nmax as the proxy of the pycnocline.'', (L235-238)

3- I think it may be very useful to have a visual illustration of the IT dynamics in this region perhaps even highlight the distinct seasonal regimes. An example is attempted bellow for Oct. 13 th 2015. There are other for the two contrasting seasons.

We have made different movies showing the propagation of internal tides during the two seasons. If necessary, we could add them to the publication as supplementary material.

4- Please see also the attached pdf. (Check yellow highlights for suggestions/corrections).

4.1  There is also similar dynamical issues with a double thermocline e.g. your Fig. 4. See works by Theo Gerkema.

Gerkema's work is now cited in the introduction, see for example: '' The scattering (reflection and refraction) and horizontal ducting of the internal tide by the pycnocline depend on its strength and width, and thus on the stratification (Gerkema, 2001, 2003), (L47-49)

4.2 This figure and figure 4 are very interesting. It is very likely that together with these results it will help understand the main dynamics of the higher mode IT systems. Maximum depth and value for N for a double thermocline should be highlighted. Maybe have two panels for ASOND. Then just add a very! brief discussion about it. No more than that is needed in this reviewer's opinion.

The maximum depth and value for N for the first 50 meters of depth are presented below. The figures could be part of the appendix, if necessary. Compared to Figure 5, to the north in the plume area, the stratification is more intense in the first 50 meters in ASOND. We briefly discuss the double pycnocline in the text (L230-248) and state that we will focus on the deeper pycnocline.

[Figure]

[Figure]

Figure: Top: $N_{max}$ value (units: $s^{-1}$) during MAMJJ (a) and (c) ASOND. Bottom: Pycnocline depth (depth of $N_{max}$, units: m) during MAMJJ (d) and (d) ASOND. The $N_{max}$ value and depth were deducted from the mean potential density over each season. Dashed blacks contours are 100 m and 2000 m isobaths. $N_{max}$ was deduced within the first 50m depth.

4.3 Summarizing this in Fig. 1 would help a lot. May add to the spectra two distinct maps (rather than just one), one for each season.

Indeed, we could have made a synthesis on figure 1, but we fear that the figure is too overloaded considering the number of elements to represent. The temporal spectra of the two seasons lead to the same qualitative conclusions as figure 1b (see figure below). We judged that it was not necessary to make this distinction at this stage of the study.

[Figure]

Figure: SSH frequency spectra based on the MAMJJ (left) and ASOND (right) hourly time series of the coherent barotropic tides (SSHBT, brown), coherent baroclinic tides (SSHBC, magenta), and the residual between the full SSH and SSHBT (SSH1, blue). The brown spectrum refers to the right scale and is shifted by 2h for clarity. The spectra are averaged offshore of the 100m isobath.

4.4 Here and whenever appropriate (including in previous sections), I think this needs a little more discussion. According to Figs. 4 and 5 that is not quite the case. Stratification along the waves' typical propagation paths has two maxima between August and December. It may be misleading to assume that the IT energy (and its dynamics) is propagating in the same fashion as that of a waveguide with a single pycnocline. That does not hinder the results of the paper nor its conclusions, but it is better to clarify and discuss (very briefly!) that different dynamics are expected as reported in previous studie e.g. see the works by Theo Gerkema (https://doi.org/10.5194/npg-10-397-2003).

This comment is similar to the one reported in 2. Gerkema's work is cited in the new version of the paper.

---

## Author Comment (AC2)

**Internal tides off the Amazon shelf during two contrasted seasons: Interactions with background circulation and SSH imprints**

Michel Tchilibou[1], Ariane Koch-Larrouy[1], Simon Barbot[1], Florent Lyard[1], Yves Morel[1], Julien Jouanno[1], and Rosemary Morrow[1]

[1]LEGOS, Université de Toulouse, CNES, CNRS, IRD, UPS, Toulouse, France

Correspondence: Michel Tchilibou (michel.tchilibou@legos.obs-mip.fr)

**Reviewer 2 answers:**

We thank the reviewer for taking the time to review our manuscript. We thank the reviewer for taking the time to review our manuscript. We found the comments extremely helpful in correcting some inconsistencies and improving the writing of the paper. In the following, the reviewers' comments are in red, and our answers are in black colors.

1- The paper's English is generally good with small grammatical errors here and there and the figures are pretty and clear (although the spectral line plot is quite dense). However, I noticed more grammatical errors in the discussion section, which seemed to be less well polished. The paper is quite long, dense, and in some places descriptive. To make sure the reader stays captivated, I suggest shortening the paper and make it more focused. The title suggests the theme is about seasonal internal tide variability (very interesting topic), but it seems to be that for the variables considered it is not very strong and/or not well communicated in the paper. The authors could go more in depth in either the energy analysis or the SSH frequency wave-number analysis (I think you could make two stand-alone papers on these topics).

We understand the reviewer's concern about the message and the length of the article. For more clarity, we have revisited all parts of the article, shorten some descriptive parts, and proposed a reorganization marking the boundary between the energetic part (section 4) and the SSH part (section 5). In the energy part, the focus is more on the differences between M2 coherent baroclinic flux over the two seasons. In the SSH part, the RMS is discussed by season and by frequency band. We have noted that it is possible to go further in the analysis in terms of energy and SSH. A new study is in progress on these two aspects, we remain open to any collaboration with the reviewer if he/she wishes.

2- Regarding the energy analysis, why do the authors consider the energetics of the coherent surface and internal tides but not the incoherent internal tides? To describe the incoherent internal tides the authors use SSH in sections 3.4 and 3.5. Similar to Pickering et al (2015) and Buijsman et al (2017), the authors could have computed incoherent signals as the tidal band-passed minus the harmonically fitted time series. This allows for a better discussion on what fraction is scattered to the incoherent internal tide and what fraction is "truly" dissipated.

Our work is done within the framework of a SWOT project for which it is essential to evaluate the RMS of the SSH due to the incoherent tide, the wavelengths at which the incoherent dominates over the coherent, and the fraction of incoherent SSH compared to the coherent. These questions are the focus of section 5 dedicated to SSH.  We already discussed the RMS of the incoherent SSH and spatial scales in the previous version of the paper.  In the revised version, we provide more detail in the text (in Section 5) and in Figure 11 (former Figure 10) on the fraction of inconsistent SSH. The use of SSH is a first step in assessing the incoherent component of the internal tide. We are considering a second step with a focus on the baroclinic flux similar to Buijsman et al., (2017), as well as the dissipation. Our results on the propagation of the baroclinic flux pushed us to work in parallel on a new configuration, extended in latitude and longitude.  The development of this configuration is almost complete and diagnostics on the baroclinic flux and dissipation will be described in another study.

3- Quite a bit of text is focuses on 8 separate generation sites. I suggest the authors focus mainly on site A as that is the largest generation site and its beam is best captured by the high resolution model. On a site note, to better investigate the energetics of this beam a bigger model domain would be better. This could also be discussed in the discussion section.

In the revised version of the article, we have created an appendix in which the discussion on other sites has been reported. The model configuration has been extended in both latitude and longitude, its exploitation is in progress and will be the subject of another publication.

4- The authors show there are significant seasonal differences in mesoscale currents and stratification, suggesting that this may be important for the energetics and beam orientation. However, they show this does not have a strong impact on the conversion. I am not completely surprised because conversion generally happens between 100 and 1000 m, where N does not change much. How does this seasonality affect the propagation of the modes for example (e.g. in their energy fluxes)? The authors could do some ray tracing to entangle the mechanisms behind the refraction?

In this region, the most energetic modes are mode 1 and mode 2. For both modes, we note that the internal tide propagates more northward in MAMJJ compared to ASOND (Figure 7). The approach in this paper is mainly qualitative, which justifies that we did not resort to quantitative analyses. Ray tracing is part of the diagnostics we are currently conducting to distinguish the effects of current and stratification on the internal tide trajectory.

5- The discussion section reads like a long summary section of the results. I suggest the authors write a shorter and more focused discussion section and also a stand-alone conclusion section. It would be nice to see a discussion section that discusses the paper's findings in light of the literature and any deficiencies the model and or analysis may have (e.g., the spectral bump at 20 km).

We have made changes to the summary and discussion section: first presenting the general conclusion of the study (section 6) and then some discussed (section 7) elements of the results in relation to the literature leading to the perspectives.

6- L35. Also cite Shriver et al (2012) here.

 Reference has been taken into account

7- L41. Some more references would be justified here. Zaron and Egbert (2014), Shriver et al (2014), Buijsman et al, (2017), Ponte et al (2015), etc etc

References have been taken into account

8- L52. I believe Zaron et al and Muller et al also wrote some papers on the seasonal variability of the internal tide. Maybe include some more references here?

We added some references on the variability of the internal tide (L49-50).

9- Figure 1. Instead of contours use curves.

Not understood

10- L105. 45 and 30 degrees are relative to what (east, north)?

These degrees are azimuts, so relative to the North pole : "the azimut being larger in Jul-Dec (45°) than in Feb-May (30°)." (L565-567)

11- L114. Please define "(un)balanced motions".

Unbalanced refers to the non-geostrophic (L110)

12- L116. I do not understand "before calculating geostrophic currents".

The sentence has been reworded (L109-110).

13- L125. Here you suggest you look into the total dissipation, but the paper discusses only the energetics of the coherent internal tide.

We have noted this confusion. In the text, we now specify "dissipation of M2 coherent".

14- L108-131. The paper raises questions that are not discussed in the same order in the manuscript.

The introduction has been rewritten, the article is organized in the order of the objectives that have been presented, namely first the characterization of the internal tide and second it's signature in SSH.

15- L174. It is not clear what definitions you mean precisely.

We were referring to the definition of Kelly et al., 2010.

16- L176. What is a non-zero mode? After depth integration?

"non-zero modes" have been replaced by "baroclinic modes" for clarity.

17- L177. Can you comment how much the improvement is (1%, 10%)? This is useful information for future studies.

Unfortunately, this information is not given in % in the cited articles. However, you can see the effects on the conversion rate and the baroclinic flux.

18- L180. Can you provide some more information on your steps here? Do you solve for the eigenfunctions using spatially varying stratification? Then you fit the U eigen modes to the harmonic constants of the 3D velocity and pressure fields? This yields the modal amplitudes that you then use in the energy analysis? Or do you only use barotropic and baroclinic SSH? You can compute baroclinic SSH from the sum of the rigid lid modal surface pressures (or is this what you do)? How many modes do you fit (this is most likely limited by frequency and by vertical and horizontal resolution; see Buijsman et al, 2020)?

We solve the equation for 10 modes (including 1 barotropic and 9 baroclinic) using variable stratification, and then we project on the different variables. The steps are detailed from L176 to L184.

19- L207. Figure 2c should be Figure 1b?

We have chosen to put this altimetric SSH map close to the model SSH to facilitate direct comparison.

20- L208. "differences" Note that the altimetry is based on 20+ years of data while your model is only based on 9 months. The longer the period over which the harmonic analysis is performed the smaller the coherent amplitude (see Ansong et al, 2015 and the appendix of Buijsman et al, 2020). Hence, your comparison may not be quite an apples-to-apples comparison. In the model far field there is clearly some coherent energy that is not present in the altimetry, possibly due to time series duration?

The goal for us is not to make a point-by-point comparison but to show that the model is able to simulate the tide with reasonable amplitude and maximum close to the coastline as in the observations. In L202, we specify that the differences between model and observations are related to the length and sampling of the series.

21- L233. There is no Coriolis balanced flow near/at the equator. How does that affect AVISO and your comparison?

Specific treatment of the SSH near the equator can give access to the geostrophic current as in AVISO. In our case, we did not do it because the latitudinal extension does not allow it. Moreover, as you said so well above, the point-by-point comparison is not obvious. What interests us is that the model reproduces the seasonal variability of the current and this is the case.

22- Figure 3. Why do you not show the currents in the Aviso data?

The currents have been added (Figure 3c and 3d)

23- Figure 4. Can you explain negative N in the figure? Maybe correct for that (set N=0)?

Negative values are related to profiles that exhibit near-surface instabilities. In the treatment of the vertical modes, these negative values are corrected.

24- L258. "expected" since you do a modal analysis do you have the answer?

In Table A.2 in the appendix, there is indeed less generation of high modes in ASOND compared to MAMJJ.

25- L260. "barrier" for the total or the coherent internal tide? This causes reflections?

Li et al., (2019) do not distinguish between total and coherent, but in general, it is the coherent. Reflection is one of the possibilities.

26- L265. What are the reasons you ignore these terms?

We wanted to specify that these terms are negligible at the first order. The sentence has been replaced (L253-255)

27- Equations 1 and 2 and Table 3. I am confused here. Why is BT dissipation positive and BC dissipation negative? Both should be positive. See Kang and Fringer (2012).

   The conversion rate was presented in Figure 6 as a barotropic energy loss, thus negative and therefore the BC dissipation was considered negative. We have corrected it for the table.

28- Eq3. Z=H not z=H+eta?

Corrected

29- L357-378. The authors discuss the dissipation of the coherent internal tide in these sections. It is generally known from Zaron and Buijsman studies that the coherent dissipation includes energy loss to the incoherent internal tide and higher harmonics. However, the authors do not clearly a priori state that. Hence, it sometimes seems if they are discussion the total dissipation. The authors could focus more on determining what fraction of the coherent dissipation scatters to the nonstationary internal tides.

We have made the effort to specify in the text that we are talking about the coherent dissipation of M2. It is a proxy, we are currently conducting specific analyses of the dissipation from the new version of the model. The current study lays the basics for future work, that is why the incoherence is discussed from the SSH.

30- Section 3.4 and 3.5. Instead of focusing on the SSH, the authors could focus on the total internal tide energetics and compare that to the coherent energetics?

The study is financed by a SWOT project, hence the interest for the SSH. We have taken into account the remarks on the total internal tide energy, and this is the subject of a new study with a model with a better latitude/longitude extension.

31- Section 3.4. This is very dense description of Figure 8. I wonder if this can be either shortened or made more quantitative (e.g. include ray tracing)?

We have shortened this description as much as possible. The approach in this first study is qualitative, we decided to look at the quantitative aspects using the extended configuration in the coming study.

32- Eqs 8-9. You assume that the higher harmonics are always incoherent. Is this really true? Can you do a harmonic analysis and fit for M4, M6, etc and see what variance fraction they comprise of the total higher harmonic variance?

In figure 1b, you can see that these harmonics are very weak. We have calculated the fraction of incoherent for the supertidal (see figure below) in the same way as the tidal (Figures 11e and 11f). This confirms that the incoherent dominates for these frequencies.

[Figure]

32- L493. "justifying the incoherence ratio of more than 0.5 noted by Zaron (2017, its Figure 8)" I am not sure if this is correct. Zaron looks at the primary frequencies, while you also include the higher harmonics. Hence this is not an apples to apples comparison.

We understand the confusion, the sentence has been removed

33- L499. "components" what components precisely?

It was the coherent and incoherent internal tide at different frequencies. The section has been modified, we hope there is no more misunderstanding

34- Figure 11. Add period [hours] to the right axis of right subplot.

This has been done see Figure 12

35- Figure 12. This is a nice figure, but the colored lines are hard to distinguish. You could add more clarity by increasing the thickness for some lines and making the panels wider?

For more clarity, we have separated Figure 12 into four panels, see Figure 13

36- L559. "However, defining the transition scale from the super tidal is delicate" It is not directly clear to what the super tidal scale transitions to (super-super tidal?). Why is it relevant to discuss this transition scale?

The transition scale is important because it gives an idea of the spatial scales above which the geostrophic will be less polluted by the non-geostrophic dynamics associated with internal waves. In our study the transition scale is determined by comparing subtidal and

tidal SSH spectra, which is not the case in Savage et al., (2019) whose method may underestimate the transition scale in regions where the internal tide remains significant at scales beyond 100 km.

37- L567. What are meridian spatial scales?

Change to spatial scales

38- L665. Refer to a Figure here?

Note was taken into account

39- L677. The MAR radiates southward waves.

We have taken this into account in our discussion of southward propagations.